# Interactions of SARS-CoV-2 envelope protein with amilorides correlate with antiviral activity

**Sang Ho Park**[1], **Haley Siddiqi**[1], **Daniela V. Castro**[1], **Anna A. De Angelis**[1], **Aaron L. Oom**[2], **Charlotte A. Stoneham**[3], **Mary K. Lewinski**[3], **Alex E. Clark**[3,4], **Ben A. Croker**[5], **Aaron F. Carlin**[3], **John Guatelli**[3], **Stanley J. Opella**[1]*

1 Department of Chemistry and Biochemistry, University of California San Diego, La Jolla, California, United States of America, 2 Department of Medicine, University of California San Diego, La Jolla, California, United States of America, 3 Division of Infectious Diseases and Global Public Health, Department of Medicine, University of California San Diego School of Medicine, La Jolla, California, United States of America, 4 Department of Cellular and Molecular Medicine, University of California San Diego, La Jolla, California, United States of America, 5 Department of Pediatrics, University of California San Diego, La Jolla, California, United States of America

* sopella@ucsd.edu

**Data Availability Statement:** All backbone chemical shift assignment files are available from the http://bmrb.io database (accession number 50813.)

## Abstract

SARS-CoV-2 is the novel coronavirus that is the causative agent of COVID-19, a some-times-lethal respiratory infection responsible for a world-wide pandemic. The envelope (E) protein, one of four structural proteins encoded in the viral genome, is a 75-residue integral membrane protein whose transmembrane domain exhibits ion channel activity and whose cytoplasmic domain participates in protein-protein interactions. These activities contribute to several aspects of the viral replication-cycle, including virion assembly, budding, release, and pathogenesis. Here, we describe the structure and dynamics of full-length SARS-CoV-2 E protein in hexadecylphosphocholine micelles by NMR spectroscopy. We also character-ized its interactions with four putative ion channel inhibitors. The chemical shift index and dipolar wave plots establish that E protein consists of a long transmembrane helix (residues 8–43) and a short cytoplasmic helix (residues 53–60) connected by a complex linker that exhibits some internal mobility. The conformations of the N-terminal transmembrane domain and the C-terminal cytoplasmic domain are unaffected by truncation from the intact protein. The chemical shift perturbations of E protein spectra induced by the addition of the inhibitors demonstrate that the N-terminal region (residues 6–18) is the principal binding site. The binding affinity of the inhibitors to E protein in micelles correlates with their antiviral potency in Vero E6 cells: HMA ≈ EIPA > DMA >> Amiloride, suggesting that bulky hydro-phobic groups in the 5' position of the amiloride pyrazine ring play essential roles in binding to E protein and in antiviral activity. An N15A mutation increased the production of virus-like particles, induced significant chemical shift changes from residues in the inhibitor binding site, and abolished HMA binding, suggesting that Asn15 plays a key role in maintaining the protein conformation near the binding site. These studies provide the foundation for com-plete structure determination of E protein and for structure-based drug discovery targeting this protein.

**Funding:** This research was supported by grants to SJO from the National Institutes of Health (https://www.nih.gov) (GM122501) and P41EB002031 for the Biotechnology Resource for Molecular Imaging of Proteins at UCSD. It was also supported by the grant from the NIH (R37AI081668) to JG, a Career Award for Medical Scientists from the Burroughs Wellcome Fund (https://www.bwfund.org), the James B. Pendleton Charitable Trust and NIH grant (K08 AI130381) to AFC, as well as grants from the NIH (R01 HL124209) and the American Asthma Foundation (http://americanasthmafoundation.org) (BSF#2017176) to BAC. ALO is supported by National Institute of Allergy and Infectious Disease (https://www.niaid.nih.gov) (F31 AI141111). The funders had no role in study design, data collection and analysis, decision to publish, or preparation of the manuscript.

**Competing interests:** The authors have declared that no competing interests exist.

## Author summary

The novel coronavirus SARS-CoV-2, the causative agent of the world-wide pandemic of COVID-19, has become one of the greatest threats to human health. While rapid progress has been made in the development of vaccines, drug discovery has lagged, partly due to the lack of atomic-resolution structures of the free and drug-bound forms of the viral proteins. The SARS-CoV-2 envelope (E) protein, with its multiple activities that contribute to viral replication, is widely regarded as a potential target for COVID-19 treatment. As structural information is essential for drug discovery, we established an efficient sample preparation system for biochemical and structural studies of intact full-length SARS-CoV-2 E protein and characterized its structure and dynamics. We also characterized the interactions of amilorides with specific E protein residues and correlated this with their antiviral activity during viral replication. The binding affinity of the amilorides to E protein correlated with their antiviral potency, suggesting that E protein is indeed the likely target of their antiviral activity. We found that residue asparagine15 plays an important role in maintaining the conformation of the amiloride binding site, providing molecular guidance for the design of inhibitors targeting E protein.

## Introduction

Severe acute respiratory syndrome coronavirus 2 (SARS-CoV-2) has garnered attention as the causative agent of the disease COVID-19. It is an enveloped RNA virus classified as a beta coronavirus [1] similar to the previously studied SARS-CoV [2] and MERS-CoV [3] viruses. While rapid progress has been made in analyzing the SARS-CoV-2 genome [4] and the development of protective vaccines [5,6], the discovery of therapeutics has lagged, largely due to the lack of structures of the viral proteins and information about their specific roles in infection, replication, and propagation. Here we apply NMR spectroscopy to the envelope (E) protein, one of the structural membrane proteins of SARS-CoV-2, in order to characterize its secondary structure, drug binding site, and effects of selected single-site mutations on its structure and binding of amiloride compounds. To accomplish these goals, the results from NMR on E protein are augmented by those from virological experiments on infected cells [7] as well as the measurement of antiviral activities of amiloride compounds.

The approximately 30kb RNA genome of SARS-CoV-2 encodes for 29 proteins (www.ncbi.nlm.nih.gov/nuccore/NC_045512). The most abundant are four structural proteins, membrane (M), envelope (E), nucleocapsid (N), and spike (S), of which M, E, and S are integral membrane proteins embedded in the lipid bilayer of the viral envelope (Fig 1). Each of these proteins exists as a homo-oligomer under some experimental conditions: a dimer or dimer of dimers for M [8], a pentamer for E [9,10], and a trimer for S [11]. The biological relevance of E protein comes from its involvement in key aspects of the virus lifecycle, including infection, replication, assembly, budding, and pathogenesis [12]. Furthermore, recombinant coronaviruses lacking E protein exhibit significantly reduced viral titers, crippled viral maturation, and yield propagation incompetent progeny [13–15]. SARS-CoV-2 E protein is a hydrophobic 75-residue protein with an amino acid sequence nearly identical to that of SARS-CoV E protein (S1 Fig) [12]. Since E protein is a viral membrane-spanning miniprotein [16], a recurring question is whether it is a viroporin. Although ion-channel activity has been detected in a variety of preparations it lacks sequence homology with any of the well-established viroporins, and

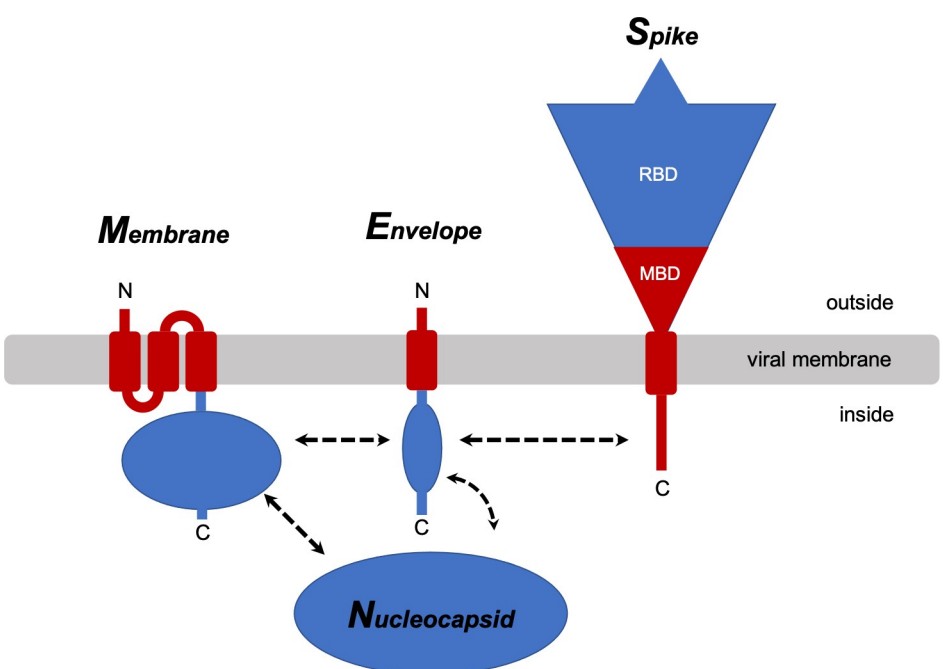

**Fig 1. Cartoon representations of the four structural proteins of SARS-CoV-2.** The membrane-associated portions of the membrane (M) protein, envelope (E) protein, and spike (S) protein are shown in red, and the extra- and intra-cellular portions are shown in blue. Proposed intraviral protein-protein interactions are indicated by the dashed arrows. RBD: receptor-binding domain; MBD: membrane-binding domain. Nucleocapsid (N) is the fourth structural protein.

there is a notable absence of charged sidechains on the interior of a pore formed by pentamers of the protein in membrane environments [9,10,17,18].

The importance of E protein for viral replication and maturation is well established, making it an attractive target for antiviral drugs. Drug design requires high-resolution structures of the protein receptor in its bound and free states. Small membrane proteins are notoriously difficult to crystallize in their native states in liquid crystalline membrane bilayers for X-ray crystallography and are too small for cryoEM to be effective. While generally suited for NMR spectroscopy, careful consideration of the membrane-like environment of the samples and the NMR experimental methods are essential [19]. Even the earliest NMR studies of membrane proteins showed that caution is called for when using micelle environments [20], because of the potential for aggregation and structural distortions [21,22]. Nonetheless, careful optimization of sample conditions has enabled solution NMR to provide valid structural information about membrane proteins that could be obtained in no other way. Moreover, we have found it essential to prepare samples of membrane proteins in micelles that yield high-resolution solution NMR spectra in order to verify that they integrate into an amphipathic membrane-like environment, are chemically pure, not mis-folded, and not aggregated before initiating significantly more demanding solid-state NMR studies of phospholipid bilayer samples. In order to ensure that solid-state NMR experiments are performed under near-native conditions, both the protein and the bilayers must be fully characterized to ensure that the protein is in its biologically active conformation and stably embedded in liquid crystalline, fully hydrated phospholipid bilayers at high lipid to protein ratios.

Here we describe solution NMR studies of full-length SARS-CoV-2 E protein and several truncated and mutated constructs in highly optimized n-hexadecylphosphocholine (HPC; fos-

choline-16) micelles. Because our novel purification scheme starts by using HPC to solubilize the protein-containing inclusion bodies and HPC is present during all subsequent steps, the polypeptides are never exposed to any other detergent or lipid, which would require exchanges, or to any organic solvent, which would require refolding. The success of the HPC-based protein purification and sample preparation scheme results in the well-resolved solution NMR spectra presented in the Figures. Furthermore, this scheme leads directly to the preparation of magnetically aligned bilayer samples that are well-suited for protein structure determination by oriented sample (OS) solid-state NMR [23].

Previous structural studies of coronavirus E protein, especially by NMR, have been simplified by using relatively short polypeptides with sequences corresponding to a substantial portion of the N-terminal domain containing the transmembrane helix that forms ion channels through homo-oligomerization as well as residues responsible for drug binding [9,10,17,18]. To date, no structural data have been presented for any full-length coronavirus E protein. Structures of a 31-residue synthetic polypeptide (residues 8–38) [17] and of a longer 58-residue expressed polypeptide (residues 8–65) containing three Cys to Ala mutations and non-native 23-residues in its N-terminus [9,18] have been described for sequences from the SARS-CoV E protein. They are highly relevant to studies of the SARS-CoV-2 E protein because the amino acid sequences of these two proteins are identical between residues 1 and 68. The partial E protein structures determined for these polypeptides in micelles by solution NMR have been modeled as pentamers [9,17,18]. In addition, an expressed 31-residue polypeptide with the same sequence as residues 8–38 of SARS-CoV-2 E protein has been studied by magic angle spinning (MAS) solid-state NMR in the presence of phospholipids, and its structure has also been modeled as a pentamer [10]. There are significant differences between the conclusions derived from these earlier studies of relatively small polypeptides missing the N-terminal seven residues and those presented here based on studies of full-length protein (residues 1–75) and two overlapping constructs encompassing the N-terminal domain (residues 1–39) and the C-terminal domain (residues 36–75). Notably, the wild-type N-terminal 39-residues are present in both the full-length and C-terminal truncated proteins.

While the transmembrane helix of E protein is thought to be largely responsible for homo-oligomerization and ion-channel activity [17,18], its highly hydrophobic nature makes modeling a channel similar to those of other miniproteins difficult [16]. Intraviral interactions between E and M proteins have been shown to involve the C-terminal domains of both proteins [24,25]. The triple cysteine motif (C40, C43, and C44) in E protein has been proposed to associate with the cysteine-rich C-terminal region of S protein by forming intermolecular disulfide bonds [26]. E protein also interacts with host proteins [12]. The C-terminal four residues, DLLV, have been identified as a PDZ-binding motif that interacts with the tight junction-associated PALS1 protein [27]. The C-terminal region of E protein that resembles the bromodomain binding site of histone H3 interacts with bromodomains 2 and 4 via acetylated Lys63, which is involved in the regulation of gene transcription [28]. These studies provide strong biological and mechanistic justification for considering coronavirus E protein as a potential drug target. Extending structural studies to samples of the full-length protein that include the complete drug binding site as well as the native N-terminus, C-terminus, and other features is essential for structure-based drug discovery. Equally important is the correlation of structural features of the protein with specific biological activities of the virus as it reproduces in human cells.

The channel activity of E protein has been suggested to play a role in viral replication [29]. A well-characterized channel blocker, hexamethylene amiloride (HMA), inhibits ion channel conductance of E proteins from HCoV-229E and MHV as well as virus replication in cultured cells [30]. HMA also inhibits the channel conductance of transmembrane-containing synthetic

and expressed polypeptides from the SARS-CoV E protein [17,18]. Although interactions of HMA with E protein of SARS-CoV have been detected in prior studies, the residues in the HMA binding site identified by NMR chemical shift perturbations varied quite a bit depending upon the specific E protein constructs and experimental conditions [9,10,17,18].

Here we characterize the secondary structure of full-length E protein from SARS-CoV-2 in HPC micelles. We also map out the complete binding sites of amiloride and three amiloride derivatives (dimethyl amiloride (DMA), ethyl isopropyl amiloride (EIPA), and HMA) and compare their binding properties. Importantly, the antiviral potency of the amiloride derivatives against SARS-CoV-2 infection of Vero E6 cells correlates well with their strength of binding as observed in the NMR experiments. The N15A and V25F mutations of the SARS-CoV-2 E protein have very different effects on the NMR spectra of the protein; the N15A mutation causes greater chemical shift perturbations over a larger region of the protein than the V25F mutation, which causes only minor changes near the site of the amino acid substitution. These mutations affect production of virus-like particle (VLP) and, in the case of N15A, the binding of HMA.

## Results

### Preparation of full-length SARS-CoV-2 E protein

In order to apply NMR spectroscopy to full-length E protein of SARS-CoV-2 it was essential to develop and implement an entirely new sample preparation scheme. We were unable to overcome the difficulties inherent in dealing with hydrophobic membrane proteins in the case of E protein using approaches that we had previously applied successfully to viral, bacterial, and human membrane proteins with between one and seven transmembrane helices [31–36]. These preparative difficulties may be among the reasons that prior NMR studies of E protein have been limited to N- and C- terminal truncated constructs with only 31 or 58 residues, which are notably missing the seven N-terminal residues that our data show to be essential components of the drug-binding site.

The ketosteroid isomerase (KSI) fusion partner facilitated the expression of high levels of three different E protein constructs, including the full-length protein (residues 1–75), as inclusion bodies in *E. coli* [32,37]. A ten-residue His-tag followed by a six-residue thrombin cleavage site, LPVRGS, inserted between the KSI and the E protein sequences enabled purification by Ni-affinity chromatography and efficient enzymatic cleavage (Fig 2C). The resulting E protein sequence differs from that of the SARS-CoV-2 isolate Wuhan-Hu-1 (NC_045512) only by the presence of two additional residues (GlySer) at the N-terminus that were originally part of the thrombin cleavage site (Fig 2B).

Our scheme for the expression and purification of the full-length wild-type and mutated, and truncated constructs of E protein is outlined in Fig 2C. They were all expressed in *E. coli* as fusion proteins and sequestered in inclusion bodies (Fig 2D). After screening many detergents informed by our extensive experience with solution NMR studies of membrane proteins [20,31,38–41] and thorough literature reviews [22], we found that the highest resolution spectra were obtained when E protein was solubilized in hexadecylphosphocholine (HPC, fos-choline-16) micelles. Chemically similar to the commonly used dodecylphosphocholine (DPC) [42], HPC has been previously considered for, but, to our knowledge, not used to study membrane proteins by NMR [43]. HPC is notable for its low critical micelle concentration (CMC, 13 μM) (www.anatrace.com). It is able to solubilize E protein and other hydrophobic membrane proteins, is effective with Ni-affinity chromatography, and at a low concentration of 0.05% w/v (1.23 mM) does not interfere with specific thrombin cleavage. This approach is highly efficient and, significantly, obviates the need for detergent exchanges or exposure to

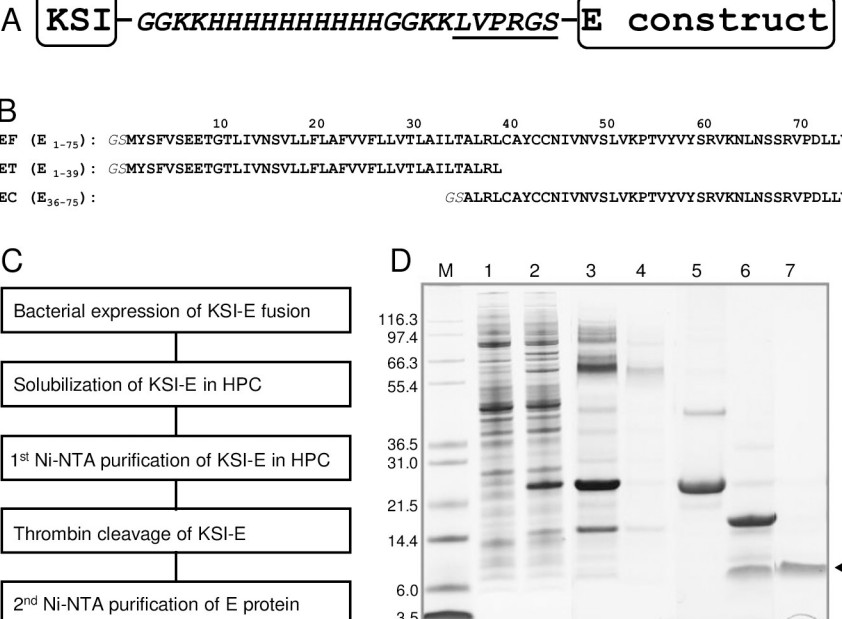

**Fig 2. Heterologous expression and purification of the full-length SARS-CoV-2 E protein and truncated protein constructs.** (A) Design of SARS-CoV-2 E protein and KSI fusion protein construct utilized for efficient bacterial expression and purification. The six residues (LVPRGS) that define the thrombin cleavage site are underlined. (B) Amino acid sequences of the polypeptides used here: full-length E protein (EF) (residues 1–75), the N-terminal transmembrane domain of E protein (ET) (residues 1–39), and the C-terminal cytoplasmic domain of E protein (EC) (residues 36–75). Two additional residues, GlySer, are present at the N-termini of all E protein constructs. (C) Block diagram of the expression and purification protocols applied to the polypeptide sequences shown in part B. (D) Example of SDS-PAGE at various stages of the expression and purification of EF: lane 1, pre-induction cells; lane 2, post-induction cells; lane 3, HPC-solubilized inclusion bodies containing the KSI-EF fusion protein; lane 4, Ni-affinity column flow through; lane 5, eluate of the KSI-EF fusion protein from the column; lane 6, after thrombin cleavage of the KSI-EF fusion protein; lane 7, arrow marks the single band of purified EF used in samples for the NMR experiments.

organic solvents at any stage of the isolation and purification process. Purified full-length E (EF) in HPC micelles runs as a monomer (~ 8.5 kDa) with a narrow band on SDS-PAGE (Fig 2D). By contrast, as observed by others [44], it runs as a broad ill-defined band on PFO (per-fluorooctanoic acid)-PAGE that may demonstrate the presence of an oligomeric species generally assumed to be a pentamer consistent with its viroporin-like properties [9,10,17,18,44,45]. All samples used in the NMR experiments were prepared directly from protein solubilized in HPC from start to finish. The resulting NMR spectra are well-resolved with narrow resonance linewidths. The samples exhibit excellent long-term stability at 50°C (Fig 3).

## Conformations of SARS-CoV-2 E protein domains are preserved

Fig 3 compares $^1$H/$^{15}$N HSQC spectra of three E protein constructs in HPC micelles. The spectra are well-resolved despite the relatively narrow span of $^1$H amide chemical shift frequencies ($< 2$ ppm) consistent with the predominantly helical conformations observed previously [9,17,18]. The backbone resonances of full-length E protein have been assigned and their chemical shifts deposited in the Biological Magnetic Resonance Data Bank (accession number: 50813) (S3 Fig and S1 Table). Notably, the observation of the expected number of resonances, with no evidence of doublings or unusual line shapes from the selectively $^{15}$N-Leu and $^{15}$N-Val labeled samples (S4 Fig), where there are no ambiguities due to spectral overlap, confirms

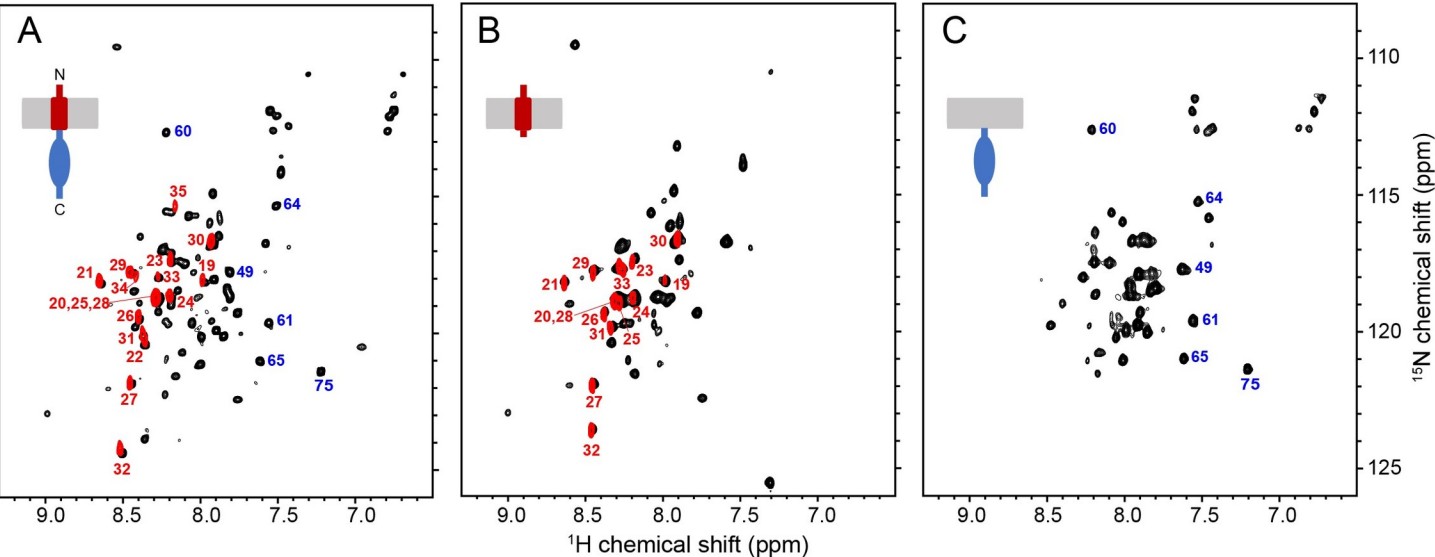

**Fig 3. Comparison of $^1H/^{15}N$ HSQC spectra of uniformly $^{15}N$-labeled E protein constructs in HPC micelles in $H_2O$ (black contours) and $D_2O$ (red contours).** (A) Full-length E protein (EF) (residues 1–75). (B) N-terminal transmembrane domain of E protein (ET) (residues 1–39). (C) C-terminal cytoplasmic domain of E protein (EC) (residues 36–75). For reference, cartoons of each construct are shown. The assignments of selected resonances are marked to distinguish among signals from ET (red numbers) and EC (blue numbers).

chemical purity and conformational homogeneity of the full-length protein in HPC under the experimental conditions. Any evidence of detergent-induced structural perturbations or heterogeneous aggregation detected in these spectra would call for further sample optimization before moving forward with solution NMR experiments or the initiation of the preparation of bilayer samples for solid-state NMR experiments.

The spectra of the N-terminal transmembrane helix-containing domain (ET) (residues 1–39) (Fig 3B) and the C-terminal cytoplasmic domain (EC) (residues 36–75) (Fig 3C) are superimposable on the spectrum of the full-length protein (EF) (residues 1–75) (Fig 3A), with the exception of signals from residues proximate to the newly formed C-terminus of ET and N-terminus of EC (S5 Fig). These results demonstrate that the folded structures of the domains are not perturbed by separation from each other, which suggests an absence of inter-domain interactions and possibly independence of their biological activities, which remains to be demonstrated *in vivo*.

Hydrogen/deuterium (H/D) exchange is an effective way to identify residues in transmembrane helices of membrane proteins [46]. When samples of the E protein constructs were prepared in >90% $D_2O$ instead of ~ 90% $H_2O$, no amide signals from residues 36–75 in the cytoplasmic domain were observable in the spectra of EF or EC; by contrast, strong signals from residues 19–35 and 19–33 were present in the spectra of EF and ET, respectively (Figs 3 and 4C), demonstrating that these residues contribute to the stable core of its unusually long trans-membrane helix. Truncation at residue 39 enhances solvent exchange at residues 34 and 35 of ET, which are 5 and 6 residues distal to its C-terminus, respectively, due to structural changes reflected in chemical shift changes of the resonances from the nine terminal residues (S4C Fig).

## Secondary structure and dynamics of full-length E protein in HPC micelles

In previous studies, evidence has been presented that the predominant secondary structure of E protein is α-helix. However, the lengths of the proposed helical segments varied widely,

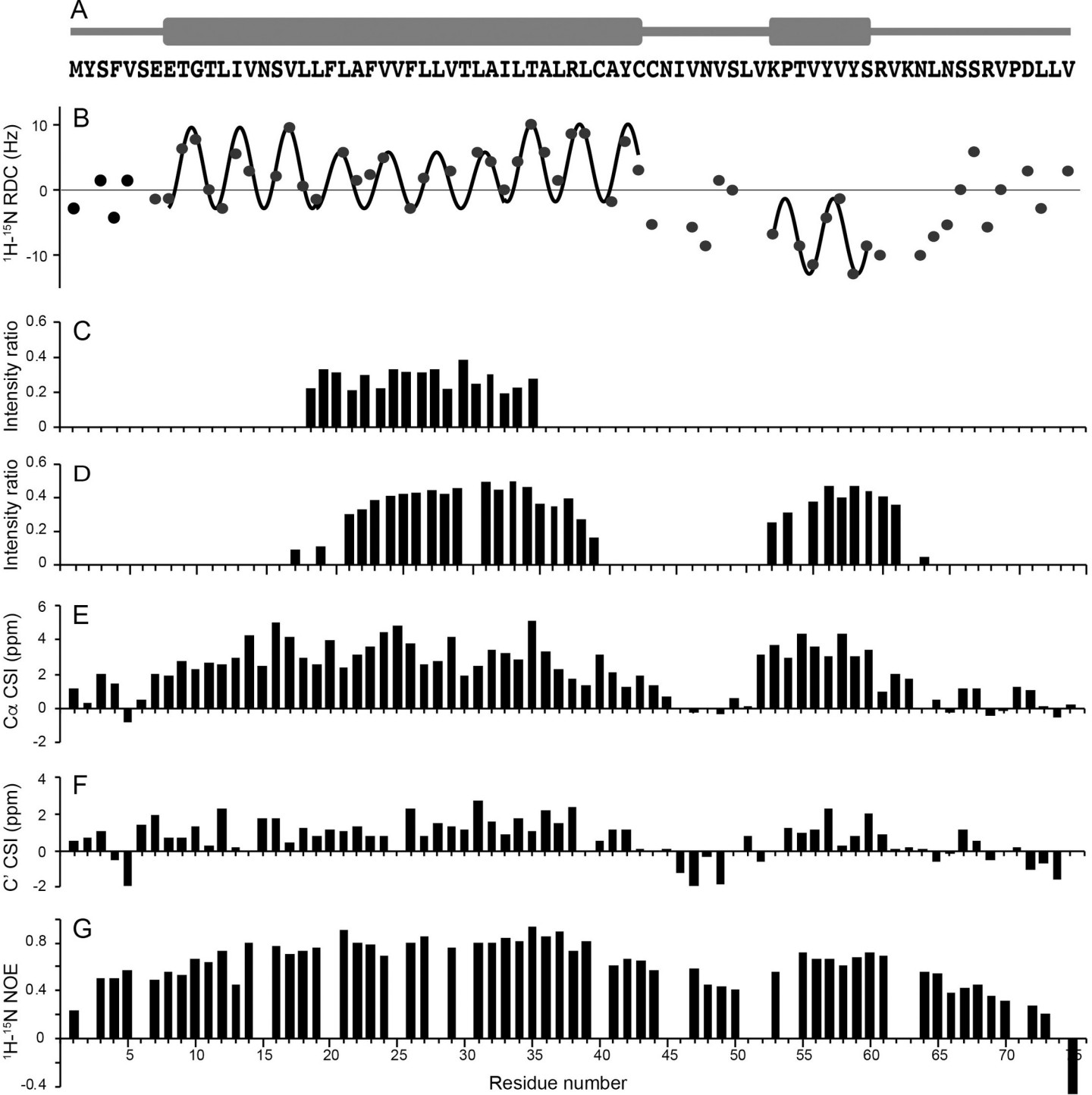

**Fig 4. Summary of NMR data obtained on full-length E protein in HPC micelles at 50°C.** (A) Schematic representation of the distribution of helical segments (thick bars) above the corresponding amino acid residues of E protein. (B) Plot of residual $^1$H-$^{15}$N residual dipolar couplings as a function of residue number. Fits to sine waves with a periodicity of 3.6 reveal the dipolar waves characteristic of alpha helical secondary structure. The residual dipolar couplings were measured on a weakly-aligned sample as shown in S6 Fig. (C) Ratios of resonance intensities with the protein in $D_2O$ compared to those in $H_2O$ solution. (D) Ratios of resonance intensities in the presence and absence of MnCl$_2$. (E and F) Chemical shift index plots of alpha (E) and carbonyl (F) carbon resonances, respectively. (G) Plot of $^1$H-$^{15}$N heteronuclear NOEs as a function of residue number.

depending upon which residues were included in the polypeptide constructs, the types of samples, and the experimental conditions [9,10,17,18]. Here we describe the secondary structure of full-length E protein in HPC micelles by analyzing the chemical shifts of backbone $^{13}$C resonances [47] and amide $^{1}$H/$^{15}$N residual dipolar couplings (RDCs) [48,49]. Further support comes from H/D exchange, manganese titration, and heteronuclear $^{1}$H/$^{15}$N NOE measurements backed up by preliminary solid-state NMR spectra of protein-containing phospholipid bilayers. Complementary results have been obtained from samples of EF, ET, and EC.

Both the $^{13}$C chemical shift index (CSI) plots (Fig 4C and 4D) and the $^{1}$H-$^{15}$N dipolar wave plot (Fig 4B) demonstrate that full-length E protein has a long 36-residue transmembrane helix and a separate short 8-residue cytoplasmic helix. None of these backbone data indicate the presence of regular secondary structure in residues 43–52 located in the region linking the two helices. Although the RDCs have significant amplitudes, as expected for a structured region, the $^{1}$H/$^{15}$N heteronuclear NOE data (Fig 4E) suggests that this well-defined internal region of the protein undergoes modest amplitude/frequency motions that are not present in the helical regions. The $^{1}$H/$^{15}$N heteronuclear NOE data also shows that residues 2–7, before the start of the N-terminal helix, and residues 61–75 following the end of the C-terminal helix exhibit gradients of increasing motion towards the termini, although even the terminal residues do not appear to be highly mobile and unstructured, as is sometimes the case in this class of proteins [32,50].

The sinusoidal waves that fit best to the magnitudes and signs of the measured RDCs as a function of residue number have a periodicity of 3.6 residues per turn, proving with a very high level of confidence that the protein has segments of regular α-helix secondary structure [48,49,51]. The addition or subtraction of a single residue at either end of the helical segments significantly degrades the quality of the fit, providing a clear demarcation of the length of the helical segments. The different average amplitudes of the two distinct dipolar waves in Fig 4B show that the two helices have different orientations relative to the direction of molecular alignment. Another notable feature is that the dipolar wave for the core region of the transmembrane helix (residues 19–34) is best fit by a sine wave with a somewhat smaller amplitude than for the rest of the long helical region (residues 8–18 and 35–43), suggesting that the 36-residue helix is not completely uniform throughout its length.

To assess the orientation of the C-terminal helix and possible interactions of the cytoplasmic domain with the hydrophilic headgroups of HPC, we examined the effects of adding paramagnetic manganese ions to samples of full-length E protein. Broadening of many $^{1}$H/$^{15}$N HSQC resonances was observed as a function of increasing the concentration of MnCl$_2$. Significantly, the signals from residues 1–16, 40–51, and 64–75 were broadened beyond detection at a concentration of 5 mM MnCl$_2$, while the signals from residues 17–39 and 52–63 remained readily observable. These signals correspond almost exactly to the residues in the core of the long hydrophobic helix (Fig 4B–4D) and the short cytoplasmic helix, with the later suggesting that the cytoplasmic helix may interact with the membrane surface.

## Interactions of SARS-CoV-2 E protein with amilorides

The chemical shift perturbations (CSPs) in the $^{1}$H/$^{15}$N HSQC spectra of the three E protein constructs (EF, ET, and EC) caused by the addition of a ten-fold molar excess of hexamethylene amiloride (HMA) to the samples are illustrated in Fig 5. The black contours represent the protein signals in the absence and the red contours in the presence of HMA. For the constructs that include the N-terminal portion of the protein, EF and ET, the chemical shifts of the corresponding residues were perturbed in the same directions and to a similar extent, as illustrated in the plots of the chemical shift changes as a function of residue number in Fig 5D and 5E. By

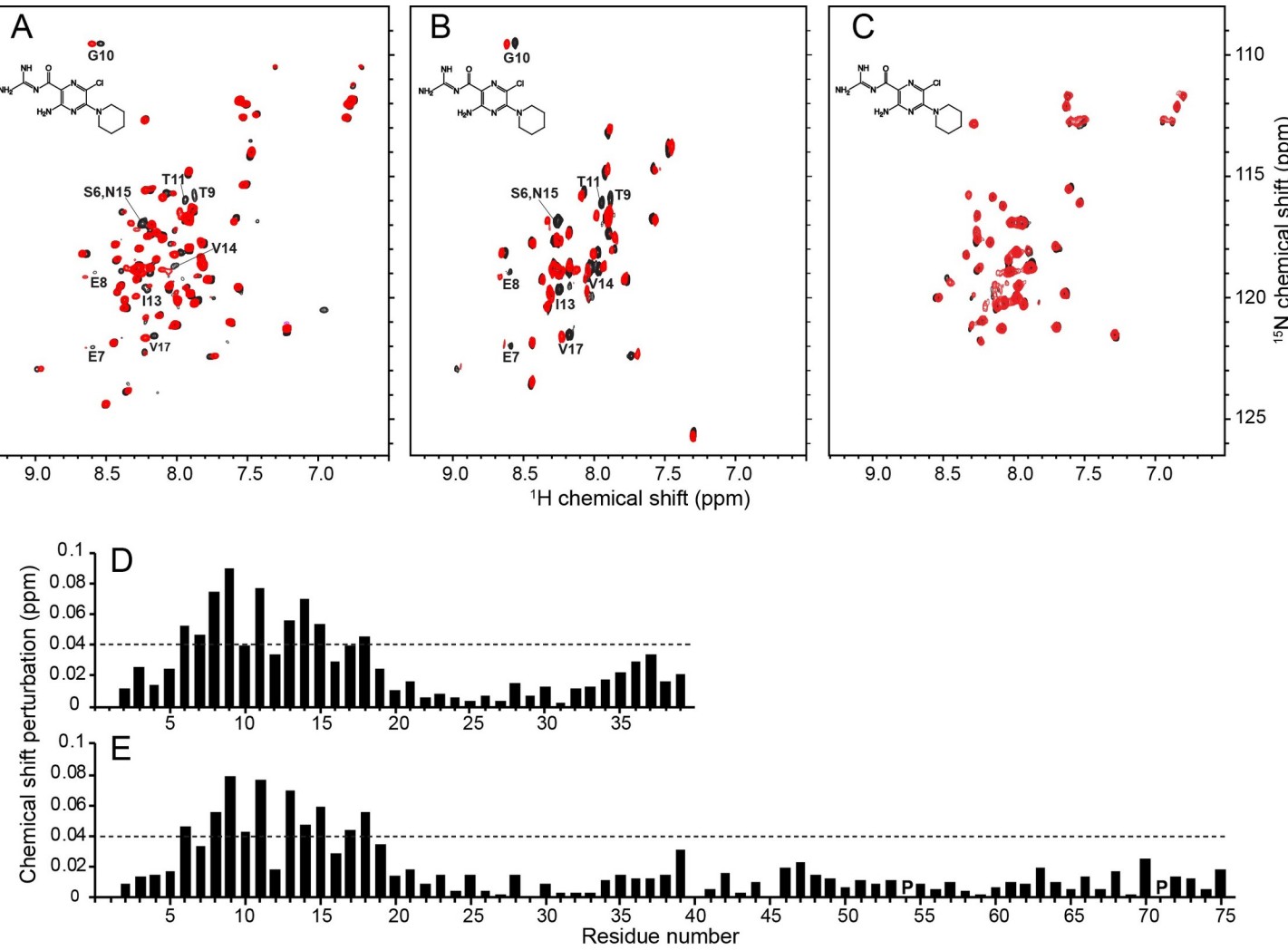

**Fig 5. Chemical shift perturbations resulting from HMA binding to E protein constructs in HPC micelles.** (A-C) Superposition of $^1$H/$^{15}$N HSQC NMR spectra of uniformly $^{15}$N-labeled E protein constructs in the absence (black contours) and presence (red contours) of HMA. (A) Full-length E protein (EF) (residues 1–75). (B) N-terminal transmembrane domain of E protein (residues 1–38) (ET). (C) C-terminal cytoplasmic domain of E protein (residues 39–75) (EC). The molar ratio of protein to HMA is 1:10. The chemical structure of HMA is shown in each spectrum. The resonances perturbed by binding HMA are labeled with their assignments. (D and E) Plots of chemical shift perturbations as a function of residue number of ET (D) and EF (E) derived from the NMR spectra in B. and A., respectively. The horizontal dotted lines represent 1.5 times the average chemical shift perturbations induced by HMA binding to ET. Proline sites are marked as "P".

contrast, no significant chemical shift changes were observed in the resonances from the cytoplasmic domain (EC) alone (Fig 5E) or as part of the full-length protein (EF) (Fig 5A). Although there is evidence that residues 2–5 are affected by drug binding, the most strongly perturbed signals are associated with residues 6–18 at the N-terminal end of the long helix and extending to the core portion distinguished by its resistance to H/D exchange and broadening by manganese ions, as well as the reduced amplitude of its dipolar wave. Qualitatively, the data in Fig 5 confirm that HMA interacts with the N-terminal domain of E protein.

The EF and ET constructs were designed to include all N-terminal residues, and these data show that the binding site definitely includes residues 6, 7, an 8, and likely residues 2, 3, 4 and 5, none of which were present in the previously studied constructs, and extends to residue 18. Nearly all of the residues that constitute the binding site belong to the highly regular helix, until it abruptly changes tilt angles at residue 18, the start of the core region. The residues

between Ser6 and Leu18 are perturbed by binding HMA and undergo facile H/D exchange: resistance to H/D exchange starts with residue 18. Notably, signals from four hydrophilic residues (Glu8, Thr9, Thr11, and Asn15) as well as Ile13 are most perturbed by HMA binding. Smaller CSPs observed in the C-terminal region of ET were not present with EF, which may be due to non-specific HMA binding to the unnatural exposed C-terminal region of ET. Titration experiments demonstrate that HMA binding occurs in fast exchange on the timescales defined by the chemical shift differences.

To compare the binding sites and affinities, we added increasing amounts of amiloride and two amiloride derivatives, dimethyl amiloride (DMA) and ethyl isopropyl amiloride (EIPA), to samples of full-length E protein (EF) and monitored their two-dimensional $^1H/^{15}N$ HSQC spectra (Fig 6). Notably, the same residues of EF were affected by all of the amiloride derivatives albeit with different magnitudes of CSPs, indicating that they all utilize the same binding site but with different binding affinities. No significant changes were observed to the EF spectrum upon addition of amiloride (Fig 6A–6D), while the largest changes were observed with EIPA (Fig 6C–6F), DMA induced moderate changes and the magnitudes of its CSPs lie between those of amiloride and HMA (Fig 6B–6E). The magnitudes of the CSPs indicate that the order of binding affinities to E protein is EIPA $\approx$ HMA > DMA >> amiloride.

## Antiviral activity of amilorides against SARS-CoV-2

The amiloride derivatives were tested for their ability to inhibit replication of SARS-CoV-2 in Vero E6 cells. Mirroring the NMR binding data of E protein in Figs 5 and 6, the compounds with bulkier aliphatic or aromatic substituents at the 5' pyrazine ring (EIPA and HMA) showed the strongest inhibition, with sub-micromolar $IC_{50}$ values, while the compounds with smaller substituents were less effective inhibitors (Fig 7). The similarity of the trends for inhibition of replication and of binding to E protein suggests that this protein may very well be a target for the antiviral activity of amiloride compounds. Of note, the most active compound examined here, HMA, shows considerable cytotoxicity (therapeutic index = 21.23), therefore, EIPA may be a better choice for potential therapeutic use (therapeutic index = 84.83) or as a starting point for further drug development.

In order to identify the stage of the viral replication cycle affected by the amiloride compounds, their antiviral activity was reevaluated at a high multiplicity of infection (MOI of 1.0 infectious units per cell) and after a relatively brief incubation (Fig 8, 18 hours). The same ranking of antiviral activity among the compounds was observed, with HMA and EIPA the most active. The observed $IC_{50}$ values were higher than those measured in the experiments summarized by the data shown in Fig 7. This was not unexpected since the antiviral assay of Fig 8 was done at an MOI ten-fold higher than that of Fig 7, and the time of incubation allowed for only one or two replication cycles (18 hours in Fig 8 compared to 48 hours in Fig 7). We observed microscopically that EIPA and HMA decreased the number of cells in each infected focus in the monolayer (Fig 8C and 8D). This effect was especially striking for HMA; most foci contained only one or two cells, suggesting that the spread of infection to adjacent cells in the foci was inhibited. To determine whether the amiloride compounds were inhibiting only this cell-cell spread or were also affecting the infectivity of the inoculum, we enumerated both the number of infected cells and the number of infected-cell-foci (containing one or more cells) and compared the $IC_{50}$ values obtained with each (Fig 8). The $IC_{50}$ values obtained using the number of infected cells (Fig 8E–8H) were less than those obtained using the number of infected-cell-foci (Fig 8I–8L). These data suggest that the amiloride compounds act late in the viral replication cycle and affect the spread of virus from cell-to-cell, although they do not exclude the possibility of a modest effect on the establishment of infection by cell-free virus.

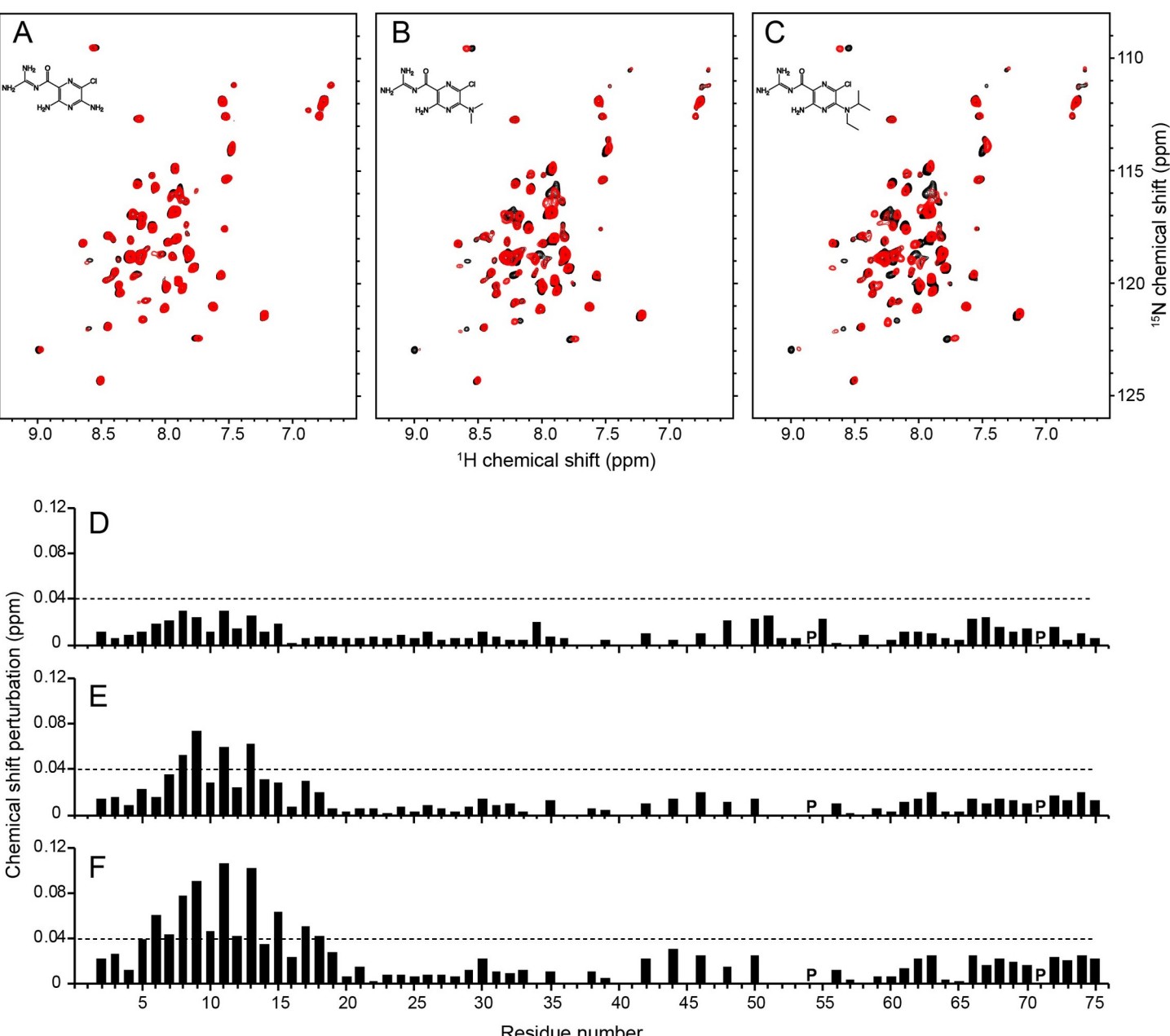

**Fig 6. Comparison of interactions of E protein with amiloride compounds.** (A-C) $^1$H/$^{15}$N HSQC NMR spectra of uniformly $^{15}$N-labeled full length E protein (EF) in the absence (black contours) and presence (red contours) of (A) amiloride, (B) DMA, and (C) EIPA. The molar ratio of EF to each compound is 1:10. Chemical structures of the drugs are shown in the spectra. (D-F) Plots of chemical shift perturbations (CSPs) in the presence of (D) amiloride, (E) DMA, and (F) EIPA as a function of residue number. The dotted lines indicate 1.5 times the average chemical shift changes of EF by EIPA. Proline sites are marked as "P".

## N15A and V25F mutations of E protein affect VLP production

Co-expression of the structural proteins M and N of SARS-CoV-2 in HEK293T cells results in a modest release of N-containing virus-like particles (VLPs) judging by the intensity of the N protein band in western blots of culture supernatants after centrifugation though a sucrose cushion. Notably, the added expression of wild-type E protein greatly stimulated the release of VLPs (Fig 9A and 9B).

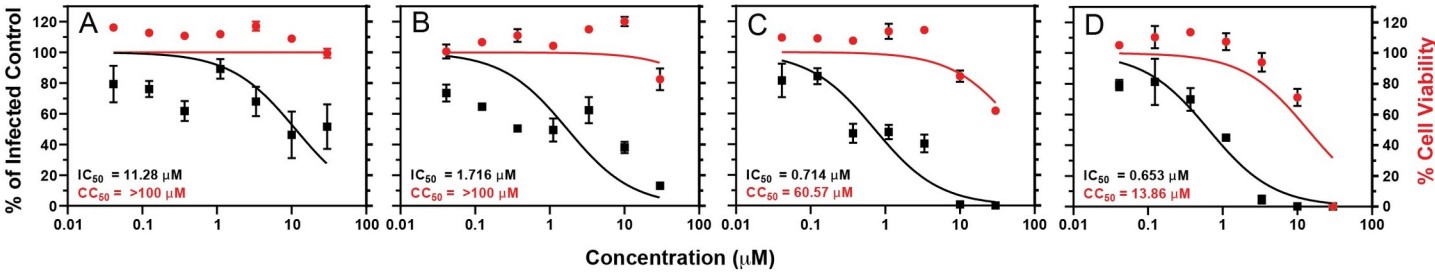

**Fig 7. Inhibition of SARS-CoV-2 infection by amiloride compounds in Vero E6 cells infected at low MOI and incubated for 48 hours.** IC$_{50}$ and CC$_{50}$ curves of (A) amiloride, (B) DMA, (C) EIPA, (D) HMA. The compounds were added at the indicated concentrations to Vero E6 cells simultaneously with authentic SARS-CoV-2 virus (MOI 0.1) and incubated for 48 hours. Inhibition of infection (solid squares and curves in black) was measured by high-content imaging for intracellular SARS-CoV-2 N protein and is relative to a DMSO-treated infected control. Cytotoxicity (solid circles and curves in red) was measured similarly using a nuclei stain and quantifying cell numbers relative to the DMSO-treated infected control. The curves were calculated using the nonlinear regression analysis in GraphPad Prism 9. IC$_{50}$ and CC$_{50}$ values for each compound are indicated in the plots.

Previous studies of E protein of Infectious Bronchitis Virus (IBV), a gamma coronavirus, showed that mutations within the transmembrane domain altered the ability of VLPs to assemble [52]. To determine the impact of similar mutations in SARS-CoV-2 E protein, two

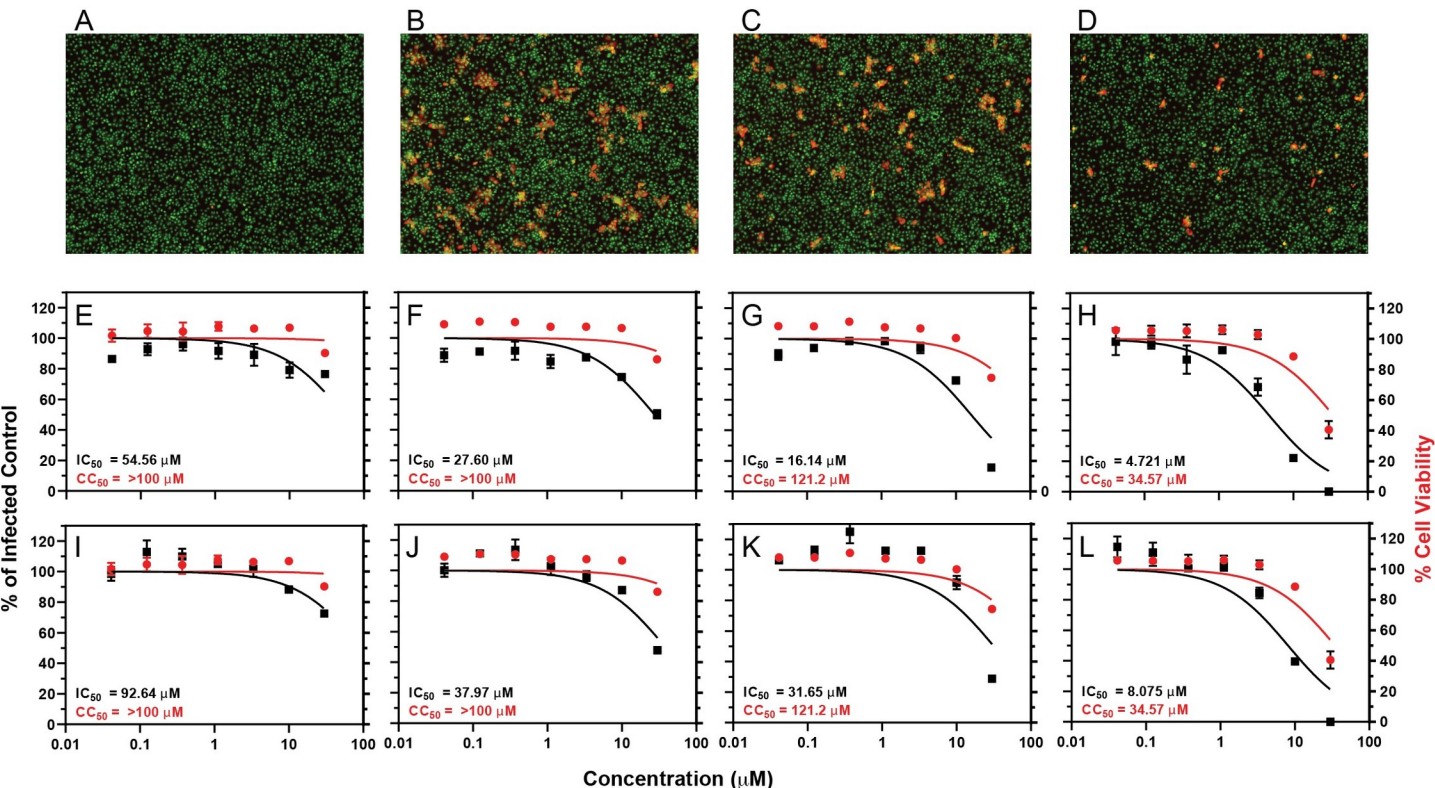

**Fig 8. Inhibition of SARS-CoV-2 infection by amiloride compounds in Vero E6 cells infected at high MOI and incubated for 18 hours.** (A-D) Images of cells. Green: nuclear stain (Sytox green); red: stain for nucleocapsid (N) using an antibody conjugated to AlexFluor594. (A) Uninfected cells. (B-D) Infected with SARS-CoV-2 and treated with (B) 0.1% DMSO, (C) 10 μM EIPA, and (D) 10 μM HMA. (E-L) IC$_{50}$ and CC$_{50}$ curves of (E and I) amiloride, (F and J) DMA, (G and K) EIPA, and (H an L) HMA for total infected cells (E-H) and foci of infection (I-L). The compounds were added at the indicated concentrations to Vero E6 cells simultaneously with SARS-CoV-2 (MOI 1) and incubated for 18 hours. Inhibition of infection (solid squares and curves in black) was measured by high-content imaging for intracellular SARS-CoV-2 N protein and is relative to a DMSO-treated infected control. Cytotoxicity (solid circles and curves in red) was measured similarly using a nuclei stain and quantifying cell numbers relative to the DMSO-treated infected control. The curves were calculated using the nonlinear regression analysis in GraphPad Prism 9. IC$_{50}$ and CC$_{50}$ values for each compound are indicated in the plots.

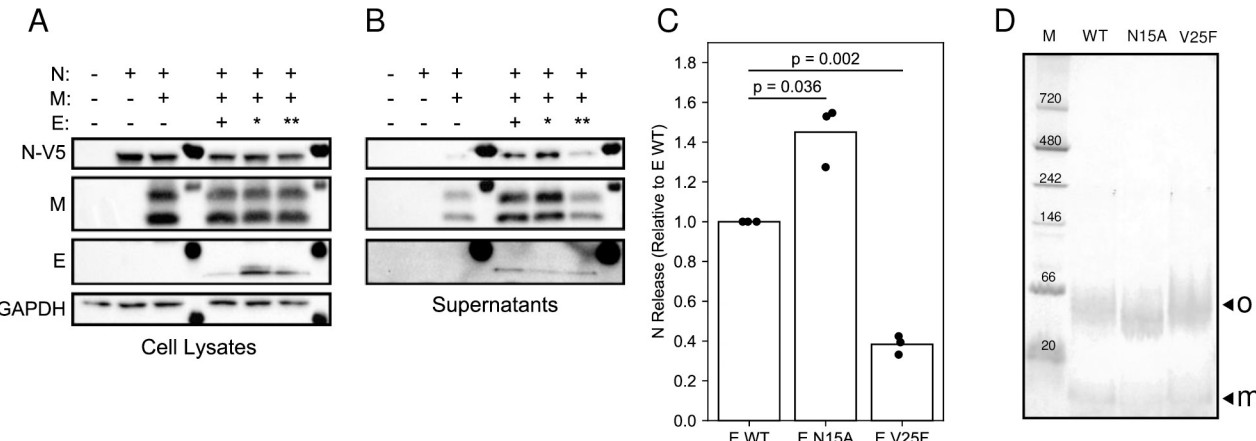

**Fig 9. Comparison of virus-like particle (VLP) production among wild-type and two mutant E proteins.** (A and B) Representative western blots of HEK293T cell lysates and sucrose cushion-purified supernatants following co-transfection with SARS-CoV-2 M, E, and N protein sequences, respectively. For E protein, * indicates the N15A mutant and ** indicates the V25F mutant. (C) Densitometry of the N protein band in purified supernatants from three independent western blot experiments. Each has M+N and the indicated E protein. The relative change over M+N without E protein is plotted for each condition. (D) PFO-PAGE of wild-type, N15A mutant, and V25F mutant E proteins. Monomer (m) and oligomer (o) bands are marked with arrows.

mutants were generated, N15A and V25F, which are analogous to IBV E protein residues Thr16 and Ala26, respectively (S1 Fig). The N15A and V25F mutations in SARS-CoV-2 E protein increased their expression compared to the wild-type protein in HEK293T cell lysates (Fig 9A). Similar to the T16A mutation in IBV E protein, the N15A mutation in SARS-CoV-2 E protein increased VLP production by approximately 40% compared to the wild-type E protein, while the V25F mutation decreased VLP production by 60% compared to wild-type E protein, similar to the effect of the A26F mutation on the IBV E protein (Fig 9B and 9C).

The mutations T16A and A26F in the IBV E protein have been shown to affect its oligomeric state [52]. However, the analogous mutations in the SARS-CoV-2 E protein do not appear to affect its oligomerization *in vitro* under our experimental conditions; in PFO-PAGE, both of these mutant E proteins ran as oligomers with only slightly different migration patterns compared to the wild-type protein (Fig 9D). As expected, both of the mutant proteins ran as monomers in SDS-PAGE with their apparent molecular weights similar to that of the wild-type protein.

## Effects of N15A and V25F mutations on structure and HMA binding of E protein

The N15A mutation results in significant chemical shift perturbations of resonances from residues throughout the N-terminal region of E protein, especially for the signals from Ser6, Glu7, Leu12, and Ser16 (Fig 10A–10C). In contrast, only minor perturbations were observed for signals from residues adjacent to the mutation site in the V25F mutant E protein (Fig 10B–10D). Since no significant differences were observed among the circular dichroism spectra from wild-type E protein and these two mutant proteins, the relatively large and wide spread chemical shift perturbations by the N15A mutation may result from changes in intermolecular hydrogen bonding involving Asn15 side chains [53].

Previous mutational studies of polypeptides containing the transmembrane helix of SARS-CoV E protein have shown that a single mutation, e.g., N15A or V25F, can disrupt ion channel activity in lipid bilayers [54,55]. We found that the N15A mutation of SARS-CoV-2 E protein decreased HMA binding, since no significant chemical shift changes are observed in the presence of HMA, with the exception of Ser6 (Fig 11A–11C). By contrast, HMA binding was not

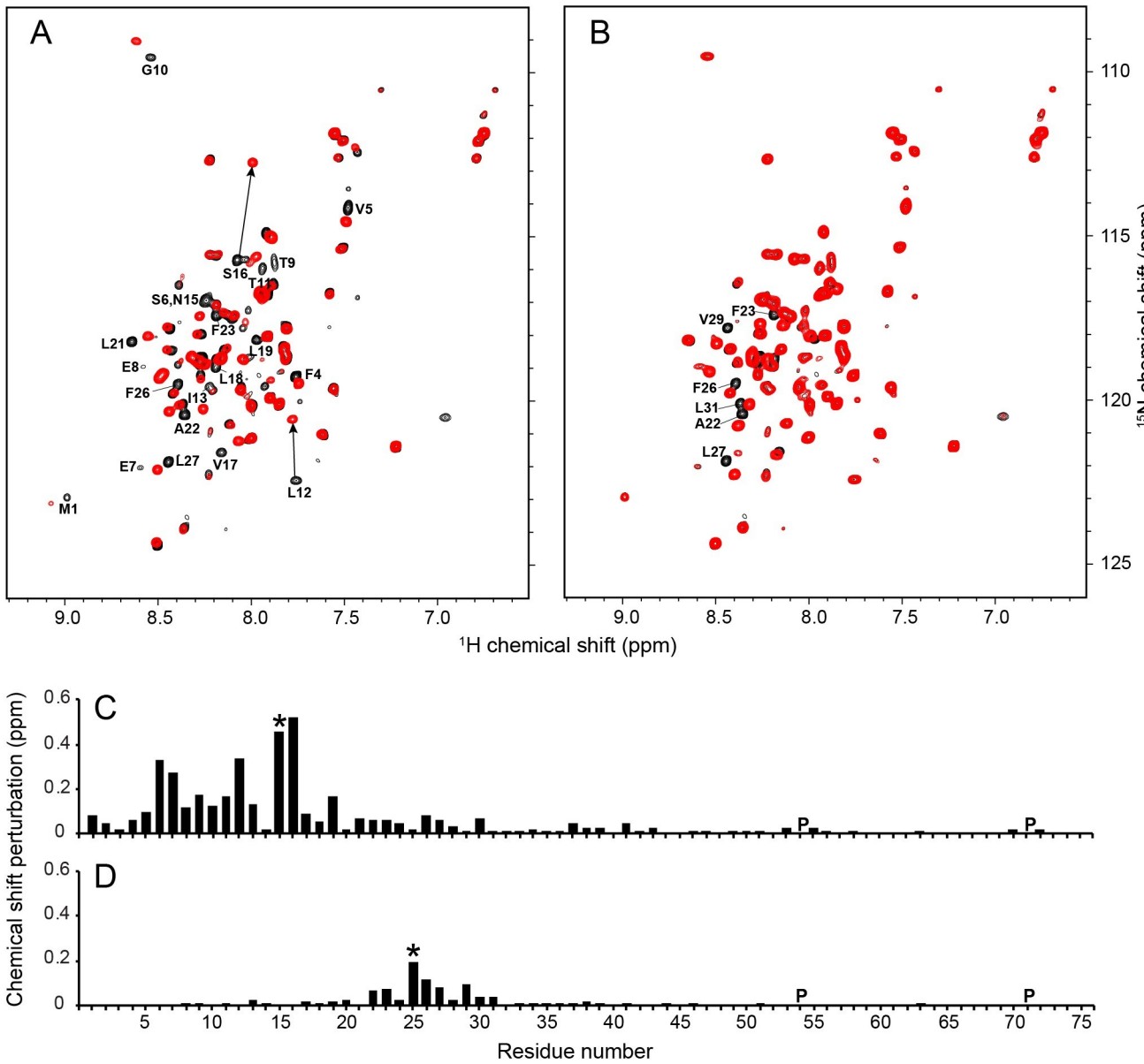

**Fig 10. Comparison of NMR data of N15A and V25F mutants of E protein. (A and B)** $^1$H/$^{15}$N HSQC NMR spectra of N15A and V25F mutant E proteins (red contours) superimposed on those from the wild-type E protein (black contours), respectively. The resonances that are significantly perturbed by the mutations are labeled with their assignments. (C and D) Chemical shift perturbation plots for the N15A and V25F mutants of E protein, respectively. The mutation sites are indicated with asterisks and the proline sites are marked as "P".

affected by the V25F mutation, since the chemical shifts of signals from residues near the HMA binding site were unchanged and their CSPs were identical to those observed for wild-type E protein (Fig 11B–11D). Based on these results, it appears that Asn15 is essential for maintaining the conformation of E protein required for binding HMA.

## Discussion

The coronavirus SARS-CoV-2 presents formidable challenges to human health, virology, and structural biology. Structural and functional studies of the envelope (E), spike (S), and

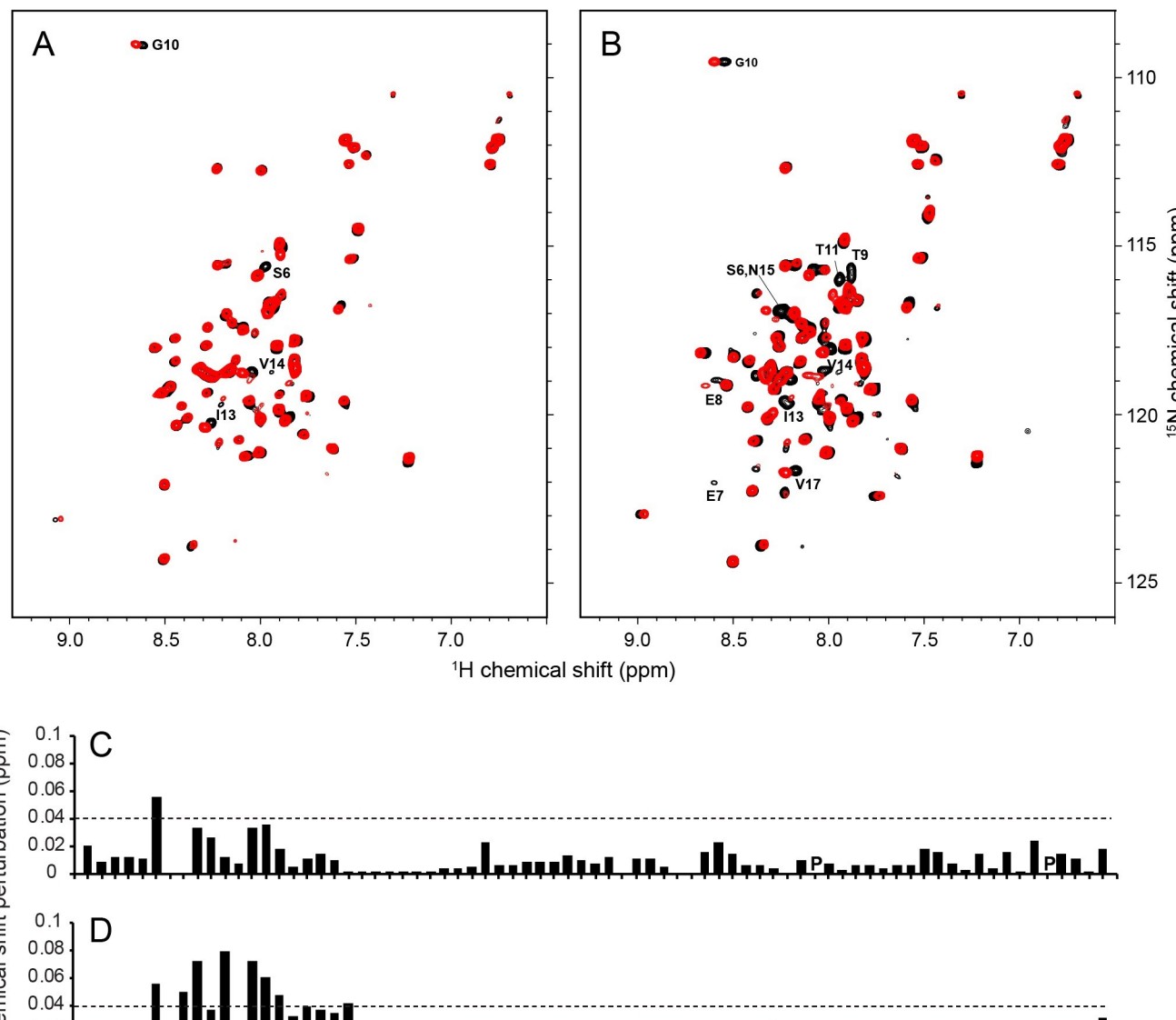

**Fig 11. Comparisons of the effects of HMA binding on the NMR spectra of N15A and V25F mutants of E protein.** (A and B) $^{1}H/^{15}N$ HSQC NMR spectra of N15A and V25F mutants of E protein in the absence (black contours) and presence (red contours) of HMA, respectively. Significantly perturbed resonances by HMA are labeled with their assignments. (C and D) Chemical shift perturbation plots of the effects of HMA binding to the N15A and V25F mutants of E protein, respectively. The dotted lines indicate 1.5 times the average chemical shift changes of V25F EF by HMA. Proline sites are marked as "P".

membrane (M) proteins are especially challenging because, as shown in Fig 1, significant portions of these proteins reside within the phospholipid bilayer of the viral envelope. Here we combine the results of solution NMR spectroscopic studies and virologic studies of SARS-CoV-2 E protein to evaluate its potential as a drug target. We focused our studies on the 75-residue full-length E protein and two overlapping truncated constructs corresponding to the N-terminal transmembrane domain (residues 1–39), that includes a long hydrophobic

helix, and the C-terminal cytoplasmic domain (residues 36–75), that includes a short helix and three cysteine residues (Fig 2B).

Heterologous expression of viral membrane proteins in *E. coli*, the most convenient system for the preparation of milligram amounts of isotopically labeled proteins, is generally problematic. Hydrophobic membrane proteins are prone to aggregation, likely from non-specific hydrophobic intermolecular interactions or possibly incorrect intra- and/or inter- molecular disulfide linkages, the latter of which is especially pertinent for SARS-CoV-2 E protein because it has three closely spaced cysteine residues in its C-terminal domain. The expression of full-length E protein from SARS-CoV, whose sequence is nearly identical to that of the 75-residue protein from SARS-CoV-2, has been reported [44]; however, a modified β-barrel construct was used as the expression tag along with urea to solubilize the inclusion bodies, followed by chemical cleavage and HPLC purification. The resulting protein in DPC or mixed DPC/SDS micelles did not yield NMR spectra suitable for structural studies. Essentially all previous NMR studies of SARS-CoV-2 E protein [9,10,17,18] were carried out on substantially smaller polypeptides with either 31 or 58 residues. In addition to the smaller number of residues in the polypeptides and the missing N- and C- terminal amino acids, the prior studies differ from those described here in several other substantial ways, including the expression system, choice of fusion protein, method of protein expression and purification, choice of micelle-forming detergent, and other experimental parameters. Not surprisingly, there are many significant differences between the findings of the previous NMR studies and those described here. Moreover, we carried out spectroscopic and virologic studies in parallel, with the results of both serving as controls, suggesting subsequent experiments and guiding the interpretation of the findings.

Nothing could be done without the preparation of isotopically labeled E protein samples suitable for NMR spectroscopy. This formidable barrier required the design and implementation of a novel bacterial expression and purification system (Fig 2). There are three notable aspects to our approach: 1) The KSI-E protein fusion protein expression system in C43(DE3) *E. coli* cells boosts expression levels and circumvents cytotoxicity by sequestering the overexpressed hydrophobic E protein in inclusion bodies; 2) Insertion of a 24-residue linker, which includes a ten-His tag and a 6-residue (LVPRGS) thrombin cleavage site, between the sequences of the KSI and E proteins, facilitates affinity chromatography purification and enzymatic cleavage because thrombin retains specificity and activity at low detergent concentrations; 3) A single "mild" detergent, HPC, with low CMC, is used to solubilize the protein throughout all steps of isolation, purification, and sample preparation. This eliminates the need for detergent or lipid exchanges and is applicable to full-length, truncated, and mutated constructs of E protein (Figs 3 and 9). Moreover, this approach to sample preparation may be generally applicable to other membrane proteins. We have already used it to prepare samples of several constructs of the membrane binding domain of the SARS-CoV-2 Spike protein (Fig 1) that yield high-resolution NMR spectra. In addition, this approach provides an expedient starting point for the preparation of samples of E protein and potentially other membrane proteins in liquid crystalline phospholipid bilayers at the high lipid to protein ratios required for solid-state NMR spectroscopy under near-native conditions. Initial comparisons between results obtained in HPC micelles by solution NMR and those obtained in phospholipid bilayers by oriented sample solid-state NMR (S7A Fig) provide assurance that the protein structure is not strongly affected by HPC. This is significant because there have been no previous NMR studies of membrane proteins in HPC micelles. The feasibility of solution NMR studies of full-length SARS-CoV-2 E protein is demonstrated by the high-resolution and signal-to-noise ratios of resonances from individual amide sites in the two-dimensional $^1$H/$^{15}$N NMR spectrum of a uniformly $^{15}$N labeled sample (Fig 3A). This includes the assignment of backbone

resonances using standard triple-resonance methods [56] on a uniformly $^{13}$C- and $^{15}$N- double-labeled sample. Following the sequential assignment of all backbone resonances in the spectra of three overlapping E protein constructs, EF (residues 1–75), ET (residues 1–39), and EC (residues 36–75), it was straightforward to characterize the overall organization, secondary structure, and local dynamics of E protein in HPC micelles using the set of experimental data aligned by residue number in Fig 4. The most striking feature to emerge is that E protein has a very long 36-residue α-helix (residues 8–43) in the N-terminal transmembrane domain. There is also a shorter 8-residue α-helix (residues 53–60) in the C-terminal domain that has a different orientation in the protein than the long helix. Since the $^{1}$H and $^{15}$N chemical shifts of the vast majority of resonances present in the two-dimensional HSQC spectra (Fig 3) of the full-length and truncated constructs overlap nearly exactly, the conformations of the N-terminal and C-terminal domains are the same whether alone or as part of the intact protein. The conservation of domain structures, also observed for the small membrane protein Vpu from HIV-1 [31,57], suggests that each domain of E protein has separate roles in the virus life cycle, although this remains to be shown in future *in vivo* experiments.

Prior NMR studies have shown E protein to be largely helical. However, the polypeptides used in the experiments and the model membrane environments differ so much that it is premature to provide a comprehensive analysis of why the lengths, locations, and distortions of the helical segments differ so drastically among various reports [9,10,17,18]. As an example, in 2009 Pervushin et al. [17] found by NMR that all 31 residues of a synthetic polypeptide with a sequence corresponding to residues 8–38 of E protein participated in a continuous α-helix in the presence of DPC. By contrast, in a 2020 report Mandala et al. [10] found by NMR a 21- or 25-residue helix, with a substantial local distortion, in the same polypeptide prepared by bacterial expression, in the presence of DMPC instead of DPC.

Here we make direct comparisons between our results on the 75-residue full-length E protein and those in the most recent report, cited above, on the widely used 31-residue doubly truncated polypeptide [10]. We find that the 36-residue transmembrane helix is quite long (residues 8–43 of the full-length E protein) compared to the more typical 21- or 25- residue transmembrane helix found by Mandala et al. The results also differ regarding the distortion of this helix. We find it to be continuous and straight, with the exception of the 17-residue core (residues 19–35), identified by resistance to H/D solvent exchange and broadening by the presence of paramagnetic $Mn^{2+}$ in the solution, and most definitively by the dipolar wave analysis that shows that this segment is also straight albeit with a detectably different tilt angle than the co-linear N- and C-terminal portions of the helix (residues 8–18 and 36–43). Instead, Mandala et al. [10] describe a singular 4-residue distortion at residues 20–23. Application of a PISA-Wheel based analysis [58,59] to oriented sample solid-state NMR data (S7A Fig) shows that the membrane-spanning helix has a large tilt angle (approx. 45°) as necessitated by hydrophobic matching with the 14-carbon methylene chains of DMPC bilayers [60]. By contrast, Mandala et al. [10] interpret their MAS solid-state NMR data to show that this helix has a very small tilt angle in the presence of DMPC. On the one hand, our solid-state NMR experiments were performed on a uniaxially aligned sample with the bilayer normal perpendicular to the direction of the magnetic field (S7A Fig), therefore the protein must be undergoing rapid rotational diffusion at 35°C in order to yield spectra with narrow single-line resonances. On the other hand, Mandala et al. [10] state that the protein does not undergo fast rigid-body uniaxial rotation at high temperatures. The differences between the structural findings in the two studies may arise from a number of possible sources, such as the difference in the lengths of the polypeptides (75 vs. 31 residues), properties of the membrane-like environments produced by HPC and DMPC, and the use of different NMR approaches (primarily solution NMR complemented by a contribution from OS solid-state NMR vs. MAS solid-state NMR). The one-

dimensional NMR spectrum in S7 Fig as well as complementary two-dimensional PISEMA spectra foreshadow the structure determination of SARS-CoV-2 E protein in phospholipid bilayers under near-native conditions.

Outstanding questions about E protein include whether it has *in vivo* ion channel activity, and whether this activity is responsible for essential biological functions. Channel activity has been observed for full-length E protein as well as N-terminal constructs containing its principal helix. This has been used as evidence that it forms a pentamer with a central pore characteristic of viroporins. Since it is small viral protein with 75 residues, it is classified as a miniprotein [16]. If its ion channel activity does indeed result from forming a defined oligomer, then it can be categorized as a viroporin. However, its primary and secondary structures differ dramatically from proteins previously described as viroporins. Most notably, the 36-residue helix of the E protein is much longer than the trans-membrane helices identified in archetypical viroporins like Vpu from HIV-1 [32] and M2 from influenza virus [61], whose shorter transmembrane helices have 18- and 25- residues, respectively.

A hallmark of viroporins is that they form homo-oligomers in the host membranes and their amphipathic transmembrane domain is essential for ion channel activity. The full-length E protein has a monomeric molecular weight of 8.5 kDa. It migrates as an oligomer with an apparent molecular weight of about 50 kDa with a minor band of monomers in PFO-PAGE (Fig 8D). The protein has three cysteines (C40, C43, and C44) and at least two of them are conserved across α/β coronaviridae (S1 Fig). The presence of reducing agents does not affect the PFO-PAGE or the NMR spectra of E protein, suggesting that cysteines are not involved in oligomerization or aggregation. Its existence in pentamers is primarily attributed to results from detergent micelle-based analytical ultracentrifugation and BN-PAGE and PFO-PAGE analysis [18,44]. IBV E protein has also been shown to exist as both monomers and oligomers during transient expression and infection by sucrose gradient analysis, and its oligomers have been proposed to correlate with stimulation of VLP production [52]. As with most other miniproteins, with the notable exception of M2 from influenza, the definition of E protein as a viroporin remains controversial.

Ion-channel activity invites the use of established channel blocking compounds as experimental probes. HMA has exhibited inhibitory activity against E protein ion channels from various coronaviruses, including MHV, HCoV-229E, SCV and FIPV, with a low micromolar range of $EC_{50}$ [30,62] as well as Vpu from HIV-1 [63] and p7 from HCV [64]. Interactions of HMA with the transmembrane domain of SARS-CoV E protein have been previously examined by NMR [9,10,17,18], and different drug binding sites have been proposed based on the chemical shift perturbations observed for different truncated E protein constructs and experimental conditions. The chemical shift perturbations we observe in spectra of 75-residue full-length SARS-CoV-2 E protein in Fig 5 provide a more complete picture of its interactions with HMA than is possible with 31- or 58- residue polypeptides. In addition, the comparison of chemical shift perturbations of three SARS-CoV-2 E protein constructs, EF, ET, and EC, in the presence of HMA clearly demonstrates that N-terminal residues 2–18 are affected by binding HMA. In our spectra, signals from hydrophilic residues (S6, E7, E8, T9, T11, and N15) are strongly affected by HMA. Minor perturbations of signals from residues in the C-terminal end of the trans-membrane helix (residues 35–37) were observed in ET but not in EF and EC and are likely due to nonspecific interactions from the truncated site. This contrasts with a prior result obtained on a truncated E protein construct with residues 8–65 that showed large CSPs for V49 and L65 [18]. A dramatic illustration that caution must be used when drug binding sites are mapped using truncated constructs is an early study of M2 [65].

A ten-fold molar excess of amiloride and its derivatives DMA, HMA, and EIPA affect resonances from the same set of amino acid residues in SARS-CoV-2 E protein, demonstrating

that they utilize the same binding site. With different CSP magnitudes, they display different binding affinities. Amiloride itself did not induce any significant changes, DMA induced modest changes, and HMA and EIPA induced the largest changes. Notably, the order of affinity of the compounds, EIPA ≈ HMA > DMA >> amiloride, correlates well with their partition coefficient (logP) values: EIPA, 1.3; HMA, 1.3; DMA, 0.1, and amiloride, -0.7 ([https://pubchem.ncbi.nlm.nih.gov](https://pubchem.ncbi.nlm.nih.gov)). Therefore, introduction of bulky aliphatic or aromatic moieties in the 5' position of the amiloride pyrazine ring, which increases the lipophilicity of the compounds, appears to increase their binding affinity for E protein. Most significantly, these findings correlate well with the antiviral activities observed for these compounds in cultures of Vero E6 cells infected with SARS-CoV-2. These data provide additional support for E protein being the likely *in vivo* target of these compounds, and they suggest that inhibition of the ion channel activity may suppress virus replication. Notably, the activity of EIPA and HMA for the Na+/K+ ATPase is higher than of amiloride [66]. Therefore, structure-based optimization of target-selectivity will be necessary in order to develop an amiloride-based drug aimed at E protein.

E protein not only stimulates viral assembly and release but also alters the secretory pathway of the cell in a manner that preserves the function of the Spike protein [67]. Consequently, while attributing the antiviral activity of the amilorides to assembly and release functions is tempting, these compounds might also impair the infectivity of virions by inhibiting the "S-preserving" function of E protein. The activities of the amilorides shown here under the conditions of high-multiplicity infection suggest that much of their antiviral action is at a late-event in the replication cycle, consistent with a block to assembly and release. Nonetheless, we have not fully excluded an effect, albeit modest, on the infectivity of cell-free virus. Such an effect would be consistent with a partial loss of S activity in mediating viral entry into target cells.

High-order oligomerization of IBV E protein has been proposed as a requirement for virus assembly [52]. Although we do not observe changes in oligomeric states of N15A or V25F mutant E proteins under our experimental conditions, comparisons of their CSPs suggests that the N15A mutation but not the V25F mutation causes a significant change in the N-terminal region. The N15A mutation affects the entire binding site and abolishes the interaction with HMA, demonstrating that residue N15 plays a key role in maintaining SARS-CoV-2 E protein's native conformation and its ability to interact with HMA. N15 (or Q15) is highly conserved in alpha and beta coronavirus E proteins. Moreover, a single Gln can mediate helix-helix associations through intermolecular hydrogen bonding within transmembrane domains [68]. Intermolecular hydrogen bonds involving the sidechain of N15 may be essential for maintaining the conformation and orientation of the N-terminal region, a conclusion also suggested by the pentameric model of the E protein oligomer [10]. The small CSPs induced by the V25F mutation are localized near the mutation site indicating that the conformation of the mutant E protein is preserved, which is consistent with its response to HMA being identical to that of the wild-type E protein.

Interest in the structure and function of SARS-CoV-2 E protein motivated the development of an efficient new approach to the expression and purification of membrane proteins so that the full-length protein could be studied. We demonstrate that HMA and EIPA bind to the N-terminal region of the E protein and exhibit antiviral activity against SARS-CoV-2. We also found that residue N15 plays an important role in maintaining the conformation of the HMA binding site, providing insight that might be helpful in the design of drugs targeting E protein. Changes associated with the N15A and V25F mutations are suggestive of involvement of E protein's N-terminal domain in virus assembly and/or release. These biological activities can be correlated with the secondary structure of E protein, which consists of a long hydrophobic transmembrane helix with a large tilt angle between residues 8–43 separated by a slightly

dynamic but still structured linker region to a second shorter helix between residues 53–60 with a significantly different tilt angle. Determination of the three-dimensional structure of E protein in phospholipid bilayers is an essential next step that should provide the structural information required to not only understand the protein's biological functions more fully, but also optimize interactions with compounds that have the potential to be developed into antiviral drugs.

## Materials and methods

### Design of SARS-CoV-2 E protein constructs

All of the studies described here utilized polypeptides with sequences based on that of the wild-type 75-residue full-length E protein from the SARS-CoV-2 isolate Wuhan-Hu-1 (NC_045512) (Fig 2B). To enhance the expression of the viral E protein in *E. coli*, a codon-optimized gene for its amino acid sequence was synthesized using the codons of highly expressed *E. coli* genes (S2 Fig) (www.idtdna.com). The codon-optimized gene was inserted into a modified pET-31b(+) vector (www.emdmillipore.com) and expressed as a ketosteroid isomerase (KSI)-fusion protein. A twenty-four-residue linker sequence incorporating a 10 His-tag and a 6-residue (LVPRGS) thrombin cleavage site was inserted between the KSI and E protein sequences (Fig 2A). The same expression and purification system was used with two truncated constructs of E protein, the N-terminal transmembrane domain (ET) (residues 1–39) and the C-terminal cytoplasmic domain (EC) (residues 36–75) (Fig 2B). Two EF mutant proteins, N15A EF and V25F EF were generated using a site-directed mutagenesis kit (www.neb.com).

### Protein expression and purification

*E. coli* strain C43(DE3) (www.lucigen.com) cells transformed with the plasmid vectors carrying the target E protein constructs were grown in minimal medium with 1 g/L ($^{15}$NH$_4$)$_2$SO$_4$ as the sole nitrogen source for producing uniformly $^{15}$N-labeled samples [69] and with 2 g/L $^{13}$C$_6$ D-glucose as the carbon source for uniformly $^{13}$C/$^{15}$N- double-labeled proteins. For selectively (by residue type) $^{15}$N-labeled samples, the minimal medium with unlabeled ammonium sulfate was supplemented with 100–500 mg/L of each of 19 amino acid residues and 100 mg/L of the $^{15}$N-labeled amino acid. The isotopically labeled compounds were obtained from Cambridge Isotope Laboratories (www.isotope.com). A preculture was grown overnight in 50 mL of Luria-Bertani (LB) broth, then a 1% (v/v) aliquot of the preculture was added to 500 mL of the minimal medium in a two-liter flask. The culture was maintained at 37˚C with shaking at 200 rpm until a cell density with an OD$_{600}$ of 0.5 was reached. Expression of the KSI-E protein fusion proteins was induced by adding isopropyl β-D-1-thiogalactopyranoside (IPTG) to a final concentration of 1 mM. After growth for 3 hr (Fig 2C lane 2) the cells were harvested by centrifugation at 5,000 xg for 20 min at 4˚C. The cell pellet was stored at -80˚C overnight.

The cell pellet was resuspended in 72 mL of a solution containing 20 mM Tris-HCl, 500 mM NaCl, pH 8 with 50 μg/mL lysozyme, and 250 units Benzonase nuclease (www.sigmaaldrich.com) per liter of culture. The cell lysate was sonicated (duty cycle 20%, output control 4, Sonic Dismembrator 550, Fisher Scientific) for 10 minutes on ice. 8 mL of 20% (v/v) Triton X-100 was added to the cell lysate to a final concentration of 2% (v/v) and incubated with gentle rotation for one hr at room temperature. The cell lysate was then centrifuged at 20,000 xg for 30 min at 4˚C. The supernatant was discarded, and the pellet containing the inclusion bodies was resuspended in 40 mL of 20 mM HEPES, 500 mM NaCl, pH 7.8. 400 mg of n-hexadecylphosphocholine (HPC, fos-choline 16, www.anatrace.com) was added to the suspension at a 1% (w/v) final concentration and Tris (2-carboxyethyl) phosphine hydrochloride (TCEP-HCl) at a final concentration of 1 mM; it was incubated with stirring for 2 hr at

room temperature or until the inclusion bodies were completely dissolved. The solubilized inclusion bodies were centrifuged at 40,000 xg for 30 min at 15˚C. The supernatant was loaded onto a Ni-NTA superflow (www.qiagen.com) column equilibrated with HPC binding buffer (0.05% HPC, 20 mM HEPES, 500 mM NaCl, pH 7.8) (Fig 2C lane 3). The column was washed with five-bed volumes of HPC binding buffer and then 10-bed volumes of HPC washing buffer (0.05% HPC, 20 mM HEPES, 500 mM NaCl, 20 mM imidazole, pH 7.8) (Fig 2C lane 4). The KSI-E protein fusion proteins were eluted with two-bed volumes of HPC elution buffer (0.05% HPC, 20 mM HEPES, 500 mM NaCl, 500 mM imidazole, pH 7.8).

The fractions containing the fusion protein were pooled and dialyzed overnight against thrombin cleavage buffer (20 mM HEPES, 50 mM NaCl, 1 mM EDTA, pH 7.8) in a 10 kDa MW cutoff dialysis membrane (www.spectrumchemical.com). Approximately 50 mg of uniformly $^{15}$N labeled KSI-E protein fusion protein was obtained from 1L of culture (Fig 2C lane 5). 10 units of high-purity thrombin (www.mpbio.com) per mg of fusion protein were added to the dialyzed solution and incubated overnight at room temperature with gentle rotation (Fig 2C lane 6). Importantly, thrombin retains its specificity and protease activity in the presence of dilute HPC. The mixture of thrombin-cleaved polypeptides was loaded onto a Ni affinity column equilibrated with HPC binding buffer and the flowthrough containing the target E protein was pooled. Typically, a yield of 10 mg of highly pure $^{15}$N-uniformly labeled E protein was obtained from one liter of cell culture (Fig 2C lane 7). The TM domain of E (ET), the cytoplasmic domain of E (EC), and the single-site mutants of EF, N15A and V25F, were all prepared following essentially the same protocol and resulted in similar yields.

## Electrophoresis

SDS-PAGE was performed using NuPAGE 4–12% Bis-Tris gels in 2-(N-morpholino)ethane sulfonic acid (MES) buffer at room temperature. The protein bands were visualized by Coomassie blue staining (Fig 2C). PFO (perfluorooctanoic acid)-PAGE was performed as previously described [32,70] using Novex 4–20% Tris-Glycine gels without SDS. The NuPAGE and Novex precast gels were obtained from Invitrogen (www.thermofisher.com). 5 μg protein samples in HPC binding buffer were mixed with the same volume of the PFO sample buffer (100 mM Tris base, 4% (w/v) NaPFO (www.alfa.com), 20% (v/v) glycerol, 0.05% bromophenol blue, pH 8.0), vortex-mixed, centrifuged for five minutes at 12,000 xg and then applied to the gel. PFO-PAGE was performed with a precooled PFO running buffer (25 mM Tris base, 192 mM glycine, 0.5% (w/v) PFO, pH 8.5) at 120 V for 3.5 hours in a cold room at 4˚C. The protein bands were visualized by Coomassie blue staining (Fig 8D).

## Sample preparation and NMR experiments

Samples for solution NMR experiments were prepared by concentrating the purified proteins with Amicon Ultra-4 10K centrifugal filters (www.endmillipore.com). Samples of 0.5 mM uniformly $^{15}$N-labeled and selectively $^{15}$N-Leu and $^{15}$N-Val labeled E protein in 5% (w/v) (123 mM) HPC, 20 mM HEPES, 50 mM NaCl, 10% (v/v) D$_2$O, 1 mM DSS, pH 6.5 were used for the two-dimensional $^1$H/$^{15}$N HSQC, $^1$H/$^{15}$N HSQC-NOESY, $^1$H/$^{15}$N heteronuclear NOE, and $^1$H/$^{15}$N IPAP-HSQC experiments [71]. Samples of 1 mM uniformly $^{13}$C,$^{15}$N-double labeled proteins in 7% (w/v) (172 mM) HPC, 20 mM HEPES, 50 mM NaCl, 10% (v/v) D$_2$O, 1 mM DSS, pH 6.5 were used for the three-dimensional HNCA, HN(CO)CA, HNCO, and HN(CA) CO experiments [56]. TCE P-HCl was added to the EF and EC samples at a final concentration of 10 mM.

The HPC concentration in the E protein ~~NMR~~ samples was estimated by comparison of the $^1$H NMR signal intensity from the HPC acyl chains with that from a 5% (w/v) (123 mM) HPC

reference sample. HPC with its low critical micelle concentration and high aggregation number did not pass through the membrane filter with a 10 kDa molecular weight cut-off. Although the HPC concentration varied slightly batch to batch after filter concentration, no significant changes in the protein NMR spectra were observed for HPC:E protein monomer molar ratios between about 100:1 and 250:1. Nevertheless, in order to maintain a consistent ratio of HPC to E protein in the samples, the HPC concentration was adjusted to 5% (w/v) for 0.5 mM E protein and 7% (w/v) for 1 mM E protein.

For H/D exchange experiments, the 90% $D_2O$ NMR samples were prepared by 9-fold dilution of the samples in 90% $H_2O$/10% $D_2O$ with a 100% $D_2O$ NMR buffer followed by concentration with a 10 kDa molecular weight cut-off filter.

$^{15}N$-$^{1}H$ residual dipolar couplings (RDCs) were measured by comparison of the $^{1}J_{NH}$ couplings of isotropic and weakly aligned EF samples. Weak alignment was induced and maintained by addition of Y21M fd bacteriophage to the protein-containing micelle solutions at a final concentration of 20 mg/mL [72].

The NMR experiments were performed on triple-resonance Bruker Avance 800 and Avance 600 spectrometers at 50˚C. The two-dimensional $^{1}H$/$^{15}N$ HSQC-NOESY data were obtained using 100 ms and 200 ms mix times. $^{1}H$/$^{15}N$ heteronuclear NOE data were obtained with a recycle delay of 4 sec. $^{1}H$ chemical shifts were referenced to 0 ppm for DSS. The NMR data were processed and analyzed using the computer programs Bruker Topspin 4 (www.bruker.com), NMRpipe/NMR Draw [73], and NMR View [74].

## Drug binding

100 mM stock solutions of amiloride, 5'-(N, N-dimethyl)-amiloride (DMA), 5-N-ethyl-N-isopropyl amiloride (EIPA), and 5-(N, N-hexamethylene)-amiloride (HMA) (www.caymanchem.com) were prepared by dissolving the appropriate amount of solid material in deuterated dimethyl sulfoxide (DMSO-$d_6$).

To observe the chemical shift perturbations of protein resonances by these compounds, two-dimensional $^{1}H$/$^{15}N$ HSQC spectra were obtained from samples containing 0.2 mM uniformly $^{15}N$-labeled protein in the absence and presence of 2 mM amiloride, DMA, HMA, or EIPA. $^{1}H$/$^{15}N$ HSQC spectra of uniformly $^{15}N$-labeled EC in the absence and presence of 2 mM HMA were also obtained. $^{1}H$/$^{15}N$ HSQC spectra of 0.2 mM uniformly $^{15}N$-labeled ET with 0, 0.25, 0.5, 1, 1.5, and 2 mM HMA present in the solution were obtained in order to track the chemical shift changes as a function of concentration. All samples used in the binding experiments contained 2% (v/v) DMSO-$d_6$ at pH 6.5 to ensure the absence of artifacts. The chemical shift perturbations were calculated using the equation CSP = $[(\Delta\delta_H)^2+(0.2\Delta\delta_N)^2]^{1/2}$, where $\Delta\delta_H$ is the change in the backbone amide $^{1}H$ chemical shift and $\Delta\delta_N$ is the change in backbone amide $^{15}N$ chemical shift of an individual resolved and assigned resonance.

## SARS-CoV-2 antiviral test of amilorides

Vero E6 and Caco-2 were obtained from ATCC and grown in DMEM (www.corning.com) with 10% FBS, 10mM HEPES, and Penicillin-Streptomycin (www.thermofisher.com). SARS-CoV-2 isolate USA-WA1/2020 (www.beiresources.org) was propagated on Caco-2 cells and infectious units quantified by focus forming assay using Vero E6 (ATCC) cells. Approximately 10e4 Vero E6 cells per well were seeded in a 96-well plate and incubated overnight. Compounds or controls were added at the indicated concentrations with addition of SARS-CoV-2 at a multiplicity of infection (MOI) equal to 1 or 0.1 as indicated in Figs 7 and 8. After incubation for 18 hr for MOI 1 or 48 hr for MOI 0.1 at 37˚C and 5% $CO_2$, the medium was removed, and the cells were incubated in 4% formaldehyde for 30 minutes at room temperature.

Formaldehyde fixed cells were washed with phosphate buffered saline and permeabilized for immunofluorescence using 0.1% Triton X-100 in PBS with 1% bovine serum albumin (BSA) fraction V (www.emdmillipore.com) and stained for SARS-CoV-2 with a primary anti-Nucle-ocapsid antibody (www.genetex.com GTX135357) labeled with AlexaFluor 594. Cells were washed twice in PBS, and the nuclei were stained with Sytox Green. Four to five images per well were obtained at 10x magnification using an Incucyte S3 (Sartorius). The percent infected cells, nuclei count, and infected foci count were calculated using built-in image analysis tools for the Incucyte S3. Foci were categorized as multi-cell foci or single infected cell by repeating the analysis with area size restrictions in the red (nucleocapsid) channel. $IC_{50}$ and $CC_{50}$ were determined using the nonlinear regression analysis in GraphPad Prism 9 with the bottom and top parameters constrained to 0 and 100, respectively. All work with authentic SARS-CoV-2 was conducted under Biosafety Level-3 conditions at the University of California San Diego. The reagent, SARS-Related Coronavirus 2, Isolate USA-WA1/2020, NR-52281 was deposited by the Centers for Disease Control and Prevention and obtained through BEI Resources, NIAID, NIH.

## VLP assays

For SARS-CoV-2 proteins, dsDNA gene fragments (gBlocks) encoding human-codon optimized sequences for M, E, and N-V5, corresponding to those of the SARS-CoV-2 Wuhan-Hu-1 isolate (genbank MN908947.3), were synthesized by Integrated DNA Technologies (www.idtdna.com). The gene fragments were inserted into the pcDNA3.1(-) plasmid backbone between the NotI and EcoRI restriction sites using In-Fusion Cloning (www.takarabio.com). The mutations in the transmembrane region of the E protein were generated from the wild-type E protein construct using the QuikChange site-directed mutagenesis kit (www.agilent.com) and verified by Sanger sequencing (www.genewiz.com). HEK293T cells were cultured in complete Dulbecco's modified Eagle medium containing 10% Fetal Bovine Serum and penicillin-streptomycin.

HEK293T cells were seeded in 6 well plates at a density of 250,000 cells/mL/well in complete Dulbecco's modified Eagle medium (DMEM). The cells were transfected the following day with 500 ng each of plasmids encoding the selected viral proteins and pcDNA2.3 plasmid backbone, using Lipofectamine 2000 (www.thermofisher.com), according to the manufacturer's protocol (3,200 ng total plasmid/well). Twenty-four hr after transfection, the supernatant from each well was clarified by centrifugation at 1,000 x$g$ for 5 min at 4°C. Clarified superna-tants were then pelleted through 20% sucrose for 1 hr at 23,500 x$g$ and 4°C. Pelleted VLPs and cells were lysed in 1X TSDS-PAGE sample buffer containing TCEP 1X Laemmli buffer with 50 mM Tris(2-carboxyethyl) phosphine (www.sigmaaldrich.com) substituted for 2-mercap-toethanol. Cell lysates were boiled for 5 min prior to use. Proteins in VLP and cell lysates were separated on 10% SDS-PAGE gels, transferred to PVDF membranes, and immunoblotted with the following antibodies (Fig 7A): mouse monoclonal anti-V5 tag (www.thermofisher.com, #R960-25), rabbit polyclonal anti-SARS M (generous gift of C. Machamer [75]), rabbit poly-clonal anti-SARS E (generous gift of C. Machamer [76]), and mouse monoclonal anti-GAPDH (www.genetex.com, #GTX627408). Primary antibodies were detected using horseradish perox-idase (HRP)-conjugated goat anti-mouse IgG (www.bio-rad.com) or HRP-donkey anti-rabbit IgG (www.bio-rad.com) and Western Clarity detection reagent (www.bio-rad.com). Apparent molecular masses were estimated using a commercial protein standard (www.thermofisher.com, PageRulePlus). Chemiluminescence was detected using a Bio-Rad Chemi Doc imaging system and analyzed using Bio-Rad Image Lab v5.1 software. Densitometry was performed using the Image Lab software (www.bio-rad.com) and statistical significance was determined with Welch's $t$-test.

## Data deposition

Backbone NMR resonances of full-length E protein was deposited in the BRMB (accession number: 50813).

## Supporting information

**S1 Fig. Sequence alignment of coronavirus E proteins.** Three subgroups are indicated. The numbers at the top of the amino acid sequence corresponds to SARS-CoV-2 E protein.
(TIF)

**S2 Fig. DNA sequence of intact full-length E protein with codons optimized for expression in *E. coli*.** The N-terminal linker sequence containing a ten histidine tag and a thrombin cleavage site is shown in italics. The sequences in bold contain the multiple restriction sites for cloning. AlwNI and XhoI sites were inserted for KSI-fusion system with pET31b(+) vector (www.emdmillipore.com). BamHI and SacI sites were inserted for GST-fusion system using pGEX-2T vector (www.sigmaaldrich.com).
(TIF)

**S3 Fig. $^1$H/$^{15}$N HSQC NMR spectrum of 0.5 mM uniformly $^{15}$N-labeled full-length E protein (residues 1–75) in 100 mM HPC at 50˚C.** The spectrum was obtained at a $^1$H resonance frequency of 800 MHz Resonance assignments are marked.
(TIF)

**S4 Fig. Comparison of $^1$H/$^{15}$N HSQC NMR spectra of uniformly $^{15}$N-labeled full-length E protein (EF) (residues 1–75) with two selectively $^{15}$N labeled E proteins.** A. $^{15}$N-Leu labeled EF. B. $^{15}$N-Val labeled EF. The spectra of selectively labeled EF (red contours) are superimposed on that of uniformly labeled EF (black contours). Resonance assignments of the selectively labeled spectra are marked. The positions of the leucine and valine residues are indicated in red in the sequence of E protein.
(TIF)

**S5 Fig. Comparison of $^1$H/$^{15}$N HSQC NMR spectra of uniformly $^{15}$N-labeled full-length E protein (EF) (residues 1–75) with two truncated constructs.** A. E protein transmembrane domain (ET) (residues 1–39). B. E protein cytoplasmic domain (EC) (residues 36–75). The spectra of the truncated constructs (red contours) are superimposed on that of the full-length E protein (EF) (black contours). C. Chemical shift perturbation plot of ET resonance frequencies compared to those of EF as a function of residue number.
(TIF)

**S6 Fig. Expanded region of $^1$H/$^{15}$N IPAP spectra of full-length E protein (EF) (residues 1–75) in HPC micelles.** A. Isotropic sample. B. Weakly aligned sample in the presence of Y21M fd bacteriophage at 20 mg/mL. Residue numbers and $^1J_{NH}$ couplings are indicated in parenthesis, respectively.
(TIF)

**S7 Fig. One-dimensional oriented sample $^{15}$N chemical shift solid-state NMR spectra.** A. Transmembrane domain of E protein (ET) from SARS-CoV-2. B. Transmembrane domain of Virus Protein U (VPU) from HIV-1 [77]. The spectrum of ET sample was obtained at 35˚C on a Bruker 900 MHz spectrometer using a home-built $^1$H/$^{15}$N double-resonance probe with a MAGC coli for the $^1$H channel and a solenoid coil for the $^{15}$N channel [78]. Uniformly $^{15}$N-labeled ET was embedded in 1,2-dimyristoyl-sn-glycero-phosphocholine (DMPC) bilayers oriented with the lipid bilayer normal perpendicular to the applied magnetic field. The molar

ratio of DMPC to ET is 395:1 and the DMPC concentration is 30% (w/v). Fast uniaxial rotational diffusion of both proteins about the bilayer normal yielded motionally averaged single line resonances. The spectra in blue are $^{15}$N chemical shift projections of the two-dimensional calculated PISA wheel spectra [58,59] with (A) a 36-residue ideal helix (PHI = -61$^o$ and PSI = -45$^o$) with its helix axis tilted 45$^o$ from the lipid bilayer normal and (B) a 18-residue ideal helix (PHI = -61$^o$ and PSI = -45$^o$) with its helix axis tilted 30$^o$ from the lipid bilayer normal. (TIF)

**S1 Table. Backbone resonance assignment and NH RDCs of full-length E protein in HPC micelles.**
(DOCX)

## Acknowledgments

We thank Francesca M. Marassi and Ye Tian for helpful discussions on the experimental results, and Xuemei Huang and Jinghua Yu for assistance with the instrumentation.

## Author Contributions

**Conceptualization:** Sang Ho Park, Aaron F. Carlin, John Guatelli, Stanley J. Opella.

**Data curation:** Sang Ho Park, Aaron L. Oom, John Guatelli, Stanley J. Opella.

**Formal analysis:** Sang Ho Park, Aaron L. Oom, Alex E. Clark, Aaron F. Carlin, John Guatelli, Stanley J. Opella.

**Funding acquisition:** Ben A. Croker, Aaron F. Carlin, John Guatelli, Stanley J. Opella.

**Investigation:** Sang Ho Park, Haley Siddiqi, Daniela V. Castro, Anna A. De Angelis, Aaron L. Oom, Charlotte A. Stoneham, Mary K. Lewinski, Alex E. Clark, Ben A. Croker, Aaron F. Carlin, John Guatelli, Stanley J. Opella.

**Methodology:** Sang Ho Park, Aaron F. Carlin, John Guatelli, Stanley J. Opella.

**Project administration:** John Guatelli, Stanley J. Opella.

**Resources:** Alex E. Clark, Ben A. Croker, Aaron F. Carlin, John Guatelli, Stanley J. Opella.

**Supervision:** Sang Ho Park, John Guatelli, Stanley J. Opella.

**Validation:** Sang Ho Park, Aaron L. Oom, Aaron F. Carlin, John Guatelli, Stanley J. Opella.

**Visualization:** Sang Ho Park, Aaron L. Oom, Aaron F. Carlin.

**Writing – original draft:** Sang Ho Park, Stanley J. Opella.

**Writing – review & editing:** Sang Ho Park, Anna A. De Angelis, Aaron L. Oom, Charlotte A. Stoneham, Mary K. Lewinski, Alex E. Clark, Ben A. Croker, Aaron F. Carlin, John Guatelli.

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
