## [Decision Letter · Decision Letter 0]

19 Apr 2021

Dear Dr. Opella,

Thank you very much for submitting your manuscript "Interactions of SARS-CoV-2 envelope protein with amilorides correlate with antiviral activity" for consideration at PLOS Pathogens. As with all papers reviewed by the journal, your manuscript was reviewed by members of the editorial board and by several independent reviewers. The reviewers appreciated the attention to an important topic. Based on the reviews, we are likely to accept this manuscript for publication, providing that you modify the manuscript according to the review recommendations.

Both reviewers were impressed with the technical advance represented by this paper. And both agree that the results are of broad significance to the community. Please address the minor points raised by both reviewers. 

Sincerely,

Benhur Lee

Section Editor

PLOS Pathogens

Benhur Lee

Section Editor

PLOS Pathogens

Kasturi Haldar

Editor-in-Chief

PLOS Pathogens

orcid.org/0000-0001-5065-158X

Michael Malim

Editor-in-Chief

PLOS Pathogens

orcid.org/0000-0002-7699-2064

Reviewer Comments (if any, and for reference):

Reviewer's Responses to Questions

**Part I - Summary**

Reviewer #1: The full-length protein is solubilized in detergent micelles for solution NMR structural characterization. This approach could be fraught with potential structural distortions from the weak modeling of the membrane environment. Solution NMR spectroscopy in nanodiscs would be much better, but more time consuming and urgency is needed. Overall this manuscript represents an unusual mix of expertise, skills, experiments and conclusions that needs to be available to the scientific community as soon as possible.

One of the primary questions from earlier work would have been to confirm the oligomerization state. While an earlier study claimed that it is a pentamer there was little evidence supporting that claim and even less evidence for claiming that the hydrophobic pore was an ion channel. Here the authors acknowledge that this is unreasonable, but do not resolve the oligomerization state, nor address the structural role of the three cysteine residues at the aqueous C-terminus of the TM helix. It seems to me that this characterization of the structure is important prior to gaining any understanding of how this protein functions. The authors claim that careful sample preparation is the key – I agree, and I do not doubt that the authors have a structured state that is uniform across their sample – but it is not clear what that state is.

The authors have done a superb job of expressing in significant quantities the E protein. The protein runs as a monomer in HPC micelles on SDS Page gels and all of the studies performed here appear to be conducted with this monomeric preparation. To think that this protein is a monomer in the virion is possible as the TM helix has only a single weakly hydrophilic residue, a threonine, in the middle of the TM helix. However, if the authors think the TM domain is much longer, such as 35 residues then do they think it is still a monomer. Of course the detergent micelle might favor a monomeric version of the protein. The N-H exchange experiments identify a hydrophobic core of residues of 14-16 residues rather typical of helices and not suggestive of an extra-long TM helix. Do I gather from the supplemental information that the authors think the TM helix would be tilted at 45° to the bilayer normal and hence a lot more embedded in a hydrocarbon environment than is likely in the micelle environment?

The RDC alignment was induced by filamentous viral particles and displays a good oscillatory behavior for a long helix. Units for the coupling should be shown. The ratios of intensity with and without D2O suggest that the TM domain of the helix is not excessively long and hence the tilt angle with the micellar axis and the presumed bilayer normal would not be large.

HMA binding is clearly to the N-terminus involving residues very close to the N-terminus up through residue 18, the start of the TM. One of the problems with using micelles is the N and C termini are not necessarily well separated, especially in a situation where there is not a lot of structure in the C-terminus. Some of the compounds with bulky substituents were shown to have sub µM IC-50’s. Both HMA and EIPA were found to be effective and apparently act late in the viral life cycle.

Co-expression of viral proteins M and N in HEK cells resulted in some viral-like particles. The number of these particles increased once E protein was also expressed.

A couple of mutations for the E protein were studied N15A and V25F – not surprisingly N15A in the N-terminus resulted in changes and V25F resulted in little change. Apparently, it had been previously demonstrated that ion channel activity by E protein had been influenced by these mutations. Since there is no way that this protein with its extremely hydrophobic TM helix is an ion channel these current results are not surprising. Apparently N15 is essential for drug binding as N15A greatly reduced drug binding.

Reviewer #2: The manuscript by Park, et al describes new structural features of the SARS-Cov-2 E(nvelope) protein and the interaction of drugs known to act as viral channel blockers. This is a well-written paper with a significant amount of experimental data: full-length E protein and two domains were studied by NMR as well as two mutants of these constructs - in addition to drug binding assessed by NMR and activity assays. Although the characterization is qualitative at this stage, substantial work is required to assign the NMR spectra of all of these species. The paper is significant in two ways:

1. It presents important new information on the structure of the E protein since previous studies used truncated constructs. Regions removed from the E protein in past studies are shown here to have both structure and influence on the protein topology. Notably, Park et al find that helix tilt angles are very different for the full-length protein. In addition, drug binding is shown to involve residues not included in past studies. Notably the NMR structural assays of these different drugs are predictive – enhanced binding contacts correlate with drug effects in activity assays.

2. Long after the COVID pandemic has resolved, this study will be cited as the first to use a novel membrane mimetic micelle – hexadecylphosphocholine (HPC). Study of the full-length protein in a detergent with an acyl chain length (C16) similar to that of a cell bilayer is a significant improvement toward approaching native environment of the cell. The quality of the NMR spectra using HCP are exceptionally good and somewhat surprising since this detergent has a very high aggregation number (178) and would be expected to form 80kD particles alone. However, it is known that membrane proteins will recruit an appropriate amount of detergent to be stabilized in a micelle (Vinogradova et al 1998 J Bio NMR 11:381). Here, Park et al show that HPC is an excellent choice when screening detergents for integral membrane protein structural studies.

**Part II – Major Issues: Key Experiments Required for Acceptance**

Reviewer #1: (No Response)

Reviewer #2: none

**Part III – Minor Issues: Editorial and Data Presentation Modifications**

Reviewer #1: It is only in the discussion that the authors mention the three Cys residues. What was done to prevent disulfide linkages between proteins?

Reviewer #2: 1. Describe method to transfer protein into 90% D2O

2. Use consistent nomenclature: D2O in text – 2H2O in methods.

3. The HPC concentration, if known accurately, may be low for this protein. The protein concentration = 500 mM (100 mM pentamer). HPC is estimated to be 123 mM or 172 mM after concentration by filtration (is this assuming that detergent does not pass through the filter? – if so, then this is an upper limit). Even without knowledge of the aggregation number of HPC containing E-protein, the ratio of pentamer to detergent monomer is 1: 1.23 – 1.72. Ideally, there should be at least one micelle per pentamer – but this ratio is unlikely under these conditions. It is possible that E-protein has adsorbed more HPC during the purification/proteolysis method. Can the authors report how the HPC concentration was determined? A direct measurement would be most useful – however, if this is not possible, simply adding detergent to an EF sample and observing if NMR signals change would be useful to demonstrate that the structural features found are not a consequence of “artifactual togetherness” owing to a limited number of micelles.

PLOS authors have the option to publish the peer review history of their article (what does this mean?). If published, this will include your full peer review and any attached files.

Reviewer #1: No

Reviewer #2: No

Figure Files:

Data Requirements:

Reproducibility:

References:

---

## [Editor Report · Decision Letter 1]

29 Apr 2021

Dear Dr. Opella,

We are pleased to inform you that your manuscript 'Interactions of SARS-CoV-2 envelope protein with amilorides correlate with antiviral activity' has been provisionally accepted for publication in PLOS Pathogens.

Best regards,

Benhur Lee

Section Editor

PLOS Pathogens

Benhur Lee

Section Editor

PLOS Pathogens

Kasturi Haldar

Editor-in-Chief

PLOS Pathogens

orcid.org/0000-0001-5065-158X

Michael Malim

Editor-in-Chief

PLOS Pathogens

orcid.org/0000-0002-7699-2064
---

## [Editor Report · Acceptance letter]

14 May 2021

Dear Dr. Opella,

We are delighted to inform you that your manuscript, "Interactions of SARS-CoV-2 envelope protein with amilorides correlate with antiviral activity," has been formally accepted for publication in PLOS Pathogens.

Best regards,

Kasturi Haldar

Editor-in-Chief

PLOS Pathogens

orcid.org/0000-0001-5065-158X

Michael Malim

Editor-in-Chief

PLOS Pathogens

orcid.org/0000-0002-7699-2064